# Highly structured, partner-sex- and subject-sex-dependent cortical responses during social facial touch

Christian L. Ebbesen [1,2,4,5]*, Evgeny Bobrov [1,2,6], Rajnish P. Rao [1] & Michael Brecht[1,3]*

Touch is a fundamental aspect of social, parental and sexual behavior. In contrast to our detailed knowledge about cortical processing of non-social touch, we still know little about how social touch impacts cortical circuits. We investigated neural activity across five frontal, motor and sensory cortical areas in rats engaging in naturalistic social facial touch. Information about social touch and the sex of the interaction partner (a biologically significant feature) is a major determinant of cortical activity. 25.3% of units were modulated during social touch and 8.3% of units displayed 'sex-touch' responses (responded differently, depending on the sex of the interaction partner). Single-unit responses were part of a structured, partner-sex- and, in some cases, subject-sex-dependent population response. Spiking neural network simulations indicate that a change in inhibitory drive might underlie these population dynamics. Our observations suggest that socio-sexual characteristics of touch (subject and partner sex) widely modulate cortical activity and need to be investigated with cellular resolution.

[1] Bernstein Center for Computational Neuroscience Berlin, Humboldt-Universität zu Berlin, 10115 Berlin, Germany. [2] Berlin School of Mind and Brain, Humboldt-Universität zu Berlin, 10115 Berlin, Germany. [3] NeuroCure Cluster of Excellence, Humboldt-Universität zu Berlin, 10115 Berlin, Germany. [4] Neuroscience Institute, New York University, New York, NY 10016, USA. [5] Skirball Institute of Biomolecular Medicine, New York University School of Medicine, New York, NY 10016, USA. [6]Present address: QUEST Center for Transforming Biomedical Research, Berlin Institute of Health (BIH), 10178 Berlin, Germany. *email: christian.ebbesen@nyumc.org; michael.brecht@bccn-berlin.de

Social touch is a powerful emotional stimulus[1]. Harnessing the ability of social touch to modulate emotion, for example, by caress or massage has emerged as a protective and therapeutic strategy for various mental health conditions, such as anxiety and depression[2]. In contrast to our detailed knowledge about cortical processing of discriminative, non-social touch, we still know very little about how social touch modulates cortical circuits.

Pioneering functional imaging studies in humans have investigated which cortical regions contribute to the top–down modulation of social-touch processing by identifying brain regions, where activity patterns depend on the social 'meaning' of the touch, rather than the actual haptic input. For example, although in fact an identical pattern of touch was always given by the same experimenter, activity in anterior cingulate, orbitofrontal, somatosensory, and insular cortices is different when subjects believe they are being touched by a man or a woman[3,4] or by a partner or a stranger[5]. This social-context-dependent modulation of cortical touch responses is negatively correlated with autism-like traits[4] and is increased by intranasal oxytocin administration[5]. In line with a role of these regions in top–down modulation of social-touch processing, somatosensory cortex, for example, is activated in a social context before any actual touch input[6,7], when observing others being touched[8], and when simply imagining pleasant or sexual touch[9].

In this paper, we apply techniques with cellular resolution in the rat, which has been pioneered in the primate by Romo and colleagues[10], namely to investigate and compare how single cortical neurons across multiple cortical areas respond during the same somatosensory stimulus. We focus on rat social facial touch, during which rats align their snouts and palpate each other's faces with their whiskers[11]. Social facial touch has attractive behavioral characteristics. First, the behavior is ecologically valid. The animals are untrained, social interactions are jointly initiated by the animals themselves, and the animals are freely moving; thus, their cortical activity presumably closely resembles activity in a natural social setting. Second, by letting animals interact with partners of both sexes, we can manipulate the social context of the touch, while keeping the actual haptic input identical or similar. In rodents, the sociosexual meaning of touching male and female conspecifics is very different[12], but during social facial interactions, male rats whisk with equal power onto conspecifics of both sexes. Females whisk onto females like a male rat would, but whisk with lower whisking amplitudes onto males[11].

Similar to humans, previous work has shown that even though whisking amplitude is lower during social facial interactions than when investigating objects, population firing rate changes[13] and membrane potential modulations[14] in rat somatosensory cortex are larger during social touch than object touch and do not correlate with whisker movements. Also similar to humans, rat somatosensory activity is modulated in a social context before actual social facial touch[14], and firing rates depend on sociosexual context, such as estrus state[13,15].

Here we ask the following questions: (i) How is information about touch and social context (partner sex) represented at the level of single neurons? (ii) How widely is this information available across five different cortical areas? (iii) How does social context impact population dynamics during touch? (iv) How does the population response structure depend on cortical area, partner sex, and sex of the subject animal itself? (v) Which cellular and network mechanisms could plausibly underlie cortical population dynamics during social touch?

We present a flexible regression approach that allowed us to investigate how social context (partner sex and subject sex) impacts cortical processing during naturalistic social touch, an untrained, self-initiated social behavior with no imposed trial structure. Touch and social context (partner sex) modulates the firing patterns of individual cortical neurons, and information about social touch and partner sex is available across cortical areas with diverse functions (somatosensory, auditory, motor, and frontal) and diverse response patterns (mostly increasing or mostly decreasing during touch). Firing rate changes of single neurons are not random across the network. On the contrary, cortical networks respond to social touch with highly structured, partner-sex and—in some cases—subject-sex-dependent population responses. Network simulations of cortical touch responses reveal that a change in inhibitory drive might underlie these population dynamics.

## Results

**Data**. In this study, we analyze the activity of 1156 neurons (single units), recorded from five cortical areas, over 7408 episodes of social facial touch in 15 female and 14 male rats (58,591 unique cell-touch pairs, averaging 51 touch episodes per cell).

**Variable behavior and neural responses during social touch**. To investigate how social touch modulates cortical processing across frontal and sensory cortices, we implanted tetrodes to record single-unit responses from freely moving, socially interacting male and female rats in the social gap paradigm (Fig. 1a). In this paradigm, rats reach across a gap between elevated platforms to engage in social facial touch, an untrained, self-initiated social behavior where the animals align their snouts and palpate each other's faces with their whiskers[11].

We recorded the activity of neurons throughout the cortical column in five cortical areas: two sensory areas, barrel cortex ('S1', the primary somatosensory representation of the mystacial vibrissae) and auditory cortex ('A1'), and three frontal areas, vibrissa motor cortex ('VMC', the primary motor representation of the mystacial vibrissae), cingulate cortex ('ACC', a putative homolog of human anterior cingulate cortex), and prelimbic cortex ('PrL', a putative homolog of human medial prefrontal cortex) (Fig. 1b). To investigate how cortical processing of social facial touch depends on the subject sex and the sex of the social interacting partner, we recorded from both male and female rats, interacting with multiple male and female conspecifics. A portion of the data analyzed here has been presented in previous studies where we investigated other questions (refs. [13,16,17], see the "Methods" section).

We plotted the activity of single neurons, across all social interactions, aligned to the onset of whisker-to-whisker touch (peri-stimulus time histograms, 'PSTHs'). From inspecting these plots, we made two preliminary observations. First, we noticed that cortical responses to social facial touch are widespread and diverse: in all brain areas, we found neurons, which both significantly increased (Fig. 1c) and significantly decreased (Fig. 1d) their activity at the onset of social facial touch. Second, we noticed that the PSTHs were highly variable from trial to trial. This variability could reflect a low correlation between firing patterns and behavior. On the other hand, if neural responses are tightly locked to behavior, but the behavior itself is highly variable, we would expect the same pattern. To discern these possibilities, we investigated the temporal statistics of the social facial touch episodes.

The median duration of a social facial touch was 1.33 s (IQR: 0.77–2.53 s, Fig. 1e, top) and the median inter-touch interval was 4.68 s (IQR: 1.24–23.78 s, Fig. 1e, middle), and both were highly variable. The duration of touch episodes varied across two orders of magnitude, the inter-touch interval varied across four orders of magnitude and the distributions of touch durations and inter-touch intervals were highly overlapping (Fig. 1e, bottom). In other

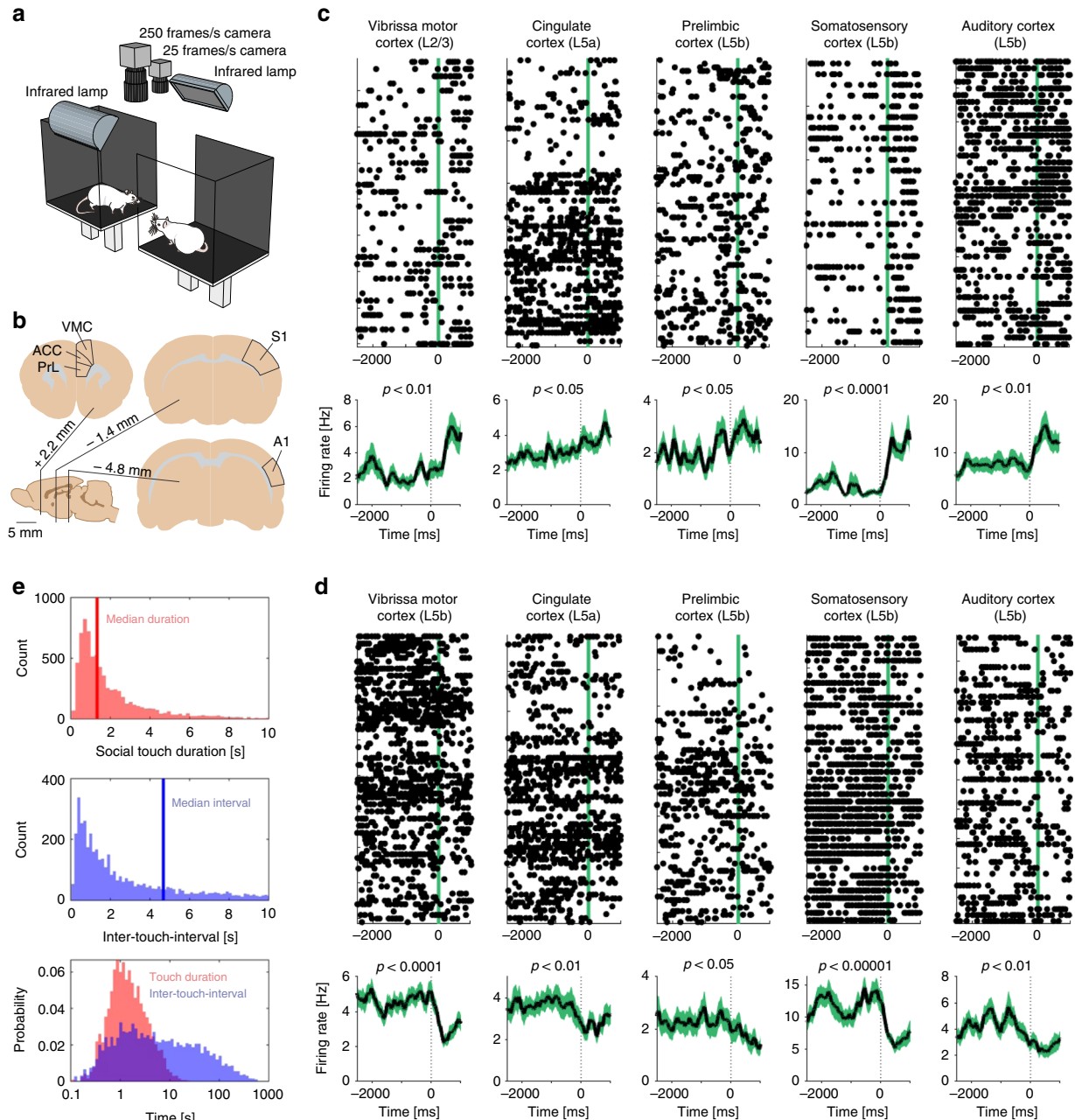

**Fig. 1** Diverse cortical responses and behavioral timing during social facial touch. **a** The social gap paradigm[11]. Rats separated on two platforms will reach across the gap to engage in social facial touch. In social facial touch, rats align their snouts and whisk to palpate each other's faces with their mystacial vibrissae. We recorded the behavior of male and female rats by videography from above in visual darkness under infrared illumination[13,16,17]. **b** Anatomical location of vibrissa motor cortex ('VMC'—the primary motor representation of the mystacial vibrissae), cingulate cortex ('ACC'—a putative homolog of human/primate anterior cingulate cortex), prelimbic cortex ('PrL'—a putative homolog of human/primate medial prefrontal cortex), barrel cortex ('S1'—the primary somatosensory representation of the mystacial vibrissae), and auditory cortex ('A1'). **c** Example peri-stimulus time histograms (PSTHs) of touch-activated single neurons (Wilcoxon signed-rank test), aligned to the first whisker-to-whisker touch in each social-touch episode. The raster plot above shows single trials, black dots indicate single spikes, and the vertical line indicates the beginning of social facial touch. The black line below indicates mean firing rate, smoothed with an Alpha kernel ($\tau = 75$ ms); shaded area indicates s.e.m. **d** Example PSTHs of touch-suppressed single neurons (Wilcoxon signed-rank test), aligned to the first whisker-to-whisker touch in each social-touch episode. Same plotting convention as in (**c**). **e** Top: Histograms of touch duration (red) and inter-touch time (blue) of social facial touch episodes. The time axes of both plots are clipped at 10 s. Below: Probability distributions of touch duration (red) and inter-touch interval (blue) of social-touch episodes, plotted on a logarithmic timescale. The two distributions span multiple orders of magnitude and strongly overlap

words, social facial touch episodes happen in bouts where the animals engage in several short touches, separated by inter-touch intervals, which are often on the same order of magnitude of the touch durations themselves. This behavioral observation suggests that a PSTH-based analysis (Fig. 1c, d) will underestimate the magnitude and overestimate the trial-to-trial variability of neuronal responses to social facial touch. The baseline period (−2000 to 0 ms) is not 'clean', so to speak, but contaminated with touch episodes, and the poststimulus period (0–500 ms) contains a mixture of trials with a wide range of touch durations.

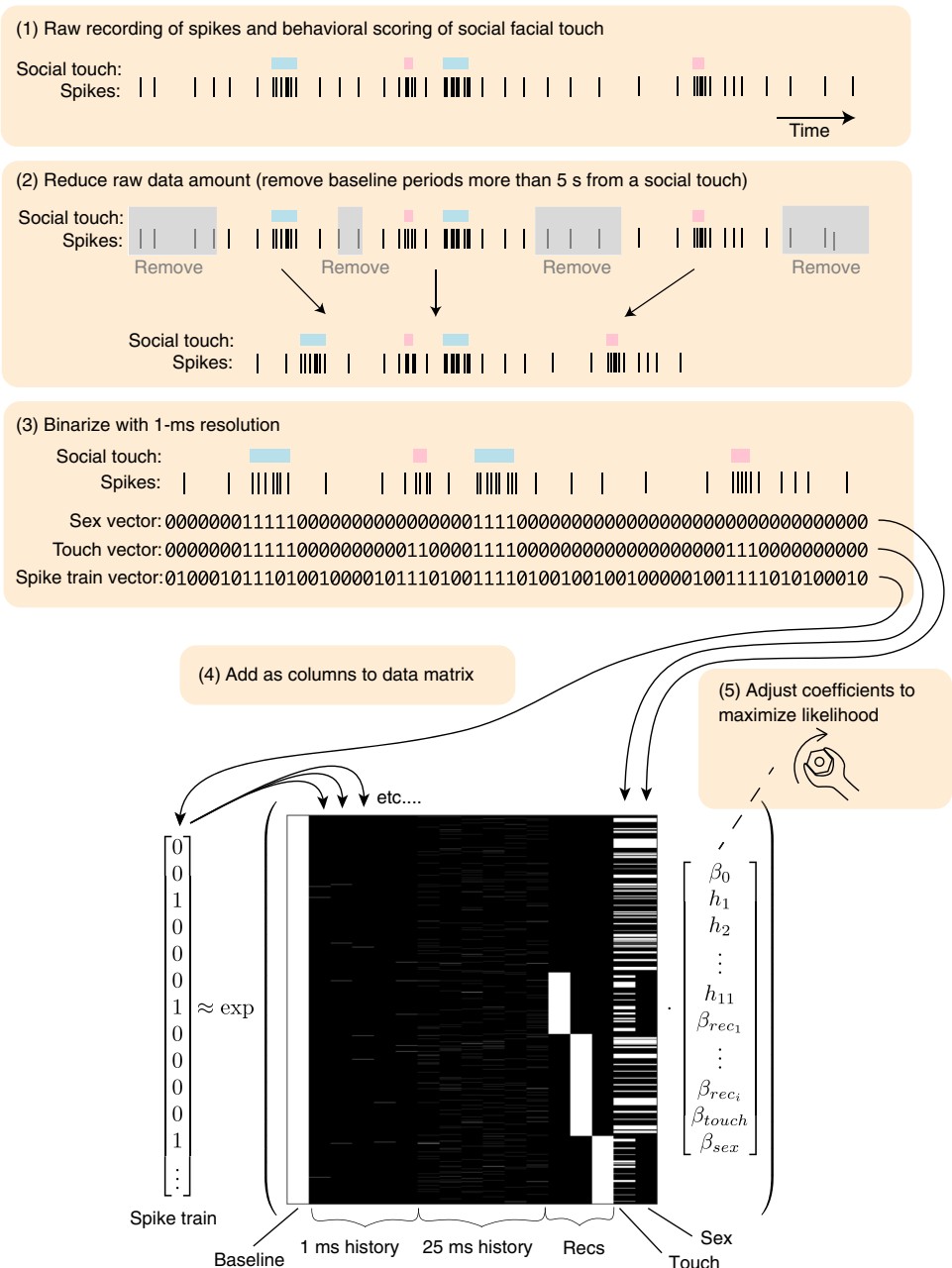

**Fig. 2** Graphical depiction of the spike train analysis. Above: Schematic depicting the five steps of the data analysis pipeline (cartoon data, with enlarged bin size for visibility; real data are binned in 1-ms bins). Below: Example predictor matrix (real data) for the generalized linear regression approach. We discretize the spike train in 1-ms bins, reduce the amount of raw data (by removing baseline periods more than five seconds from a social interaction), and model the firing rate as a Poisson process[18,19] (see Methods). The predictor matrix has the following columns (left to right): a constant baseline rate; five 1-ms spike history bins; six 25-ms history bins; three one-hot columns to model possible changes in baseline between the (in this example case) four recordings; a one-hot column indicating all social-touch episodes; a one-hot column indicating the sex of the stimulus animal (0/1 corresponds to male/female, in this example case, the interaction partner animals were male in recording one and recording three). The vector indicates regression coefficients, which we fit numerically by likelihood maximization

**Single cortical neurons signal social touch and partner sex.** To overcome the challenges presented by the highly variable temporal statistics of naturalistic social interactions, we used a generalized linear regression approach. Briefly, we modeled the spiking activity of the neurons as a Poisson process with history dependence and baseline fluctuations[18,19] (graphical depiction of the data analysis and statistical modeling 'pipeline' shown in Fig. 2). We used a maximum-likelihood approach combined with a nonparametric shuffling procedure to investigate the effect of social touch and the sex of the social interaction partner, while maintaining the

information about duration and inter-touch interval of every single touch episode (see Methods). We quantified the modulation of firing rates as a regression coefficient (a $\beta$-coefficient), which can be thought of as a modulation index that measures firing rate changes as 'fold changes' ($\beta = 1$ corresponds to an $e$-fold increase, $\beta = -1$ corresponds to an $e$-fold decrease; motivation and an in-depth introduction to the use of $\beta$-coefficients as a measure of firing rate changes across neural populations is provided in Supplementary Note 1). This approach allowed us to ask how social touch impacts neural activity, despite the behavioral variability.

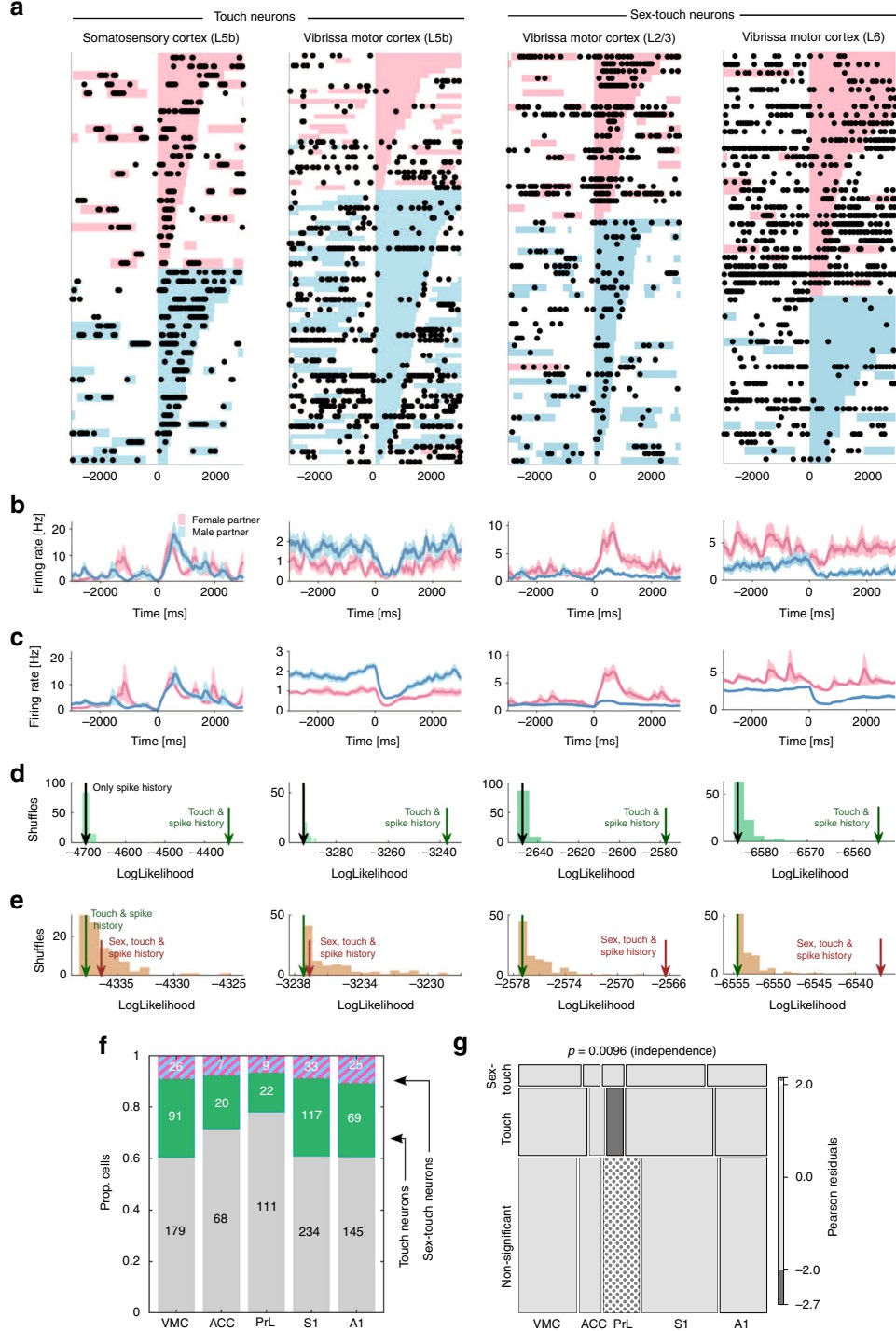

By using this statistical approach, we identified neurons that were modulated by social touch ('touch neurons'), and neurons that were modulated by social touch, but responded differently when touching male and female conspecifics ('sex-touch neurons'). Touch and sex-touch neurons had very diverse response patterns. We found both touch and sex-touch neurons that increased and decreased their firing rates during social touch (Fig. 3a–e, left), and sex-touch neurons that responded more strongly to either female or male conspecifics (Fig. 3a–e, right, additional examples of the diversity of sex-touch responses across the cortical areas are plotted in Supplementary Fig. 1a–e).

As we expected based on the temporal patterns of the behavior (Fig. 1e), a lot of the intertrial variability in the baseline period

was due to variations in behavior. For example, many of the spikes in the baseline period of the example-activated touch neuron (a layer 5b neuron in S1) and many of the pauses in firing in the example-suppressed touch neuron (a layer 5b neuron in VMC) coincided with social-touch episodes (Fig. 3a). Similarly, much of the intertrial variability in the poststimulus period was due to variations in the length of the social touch (Fig. 3a).

Across all investigated areas, we found that a large proportion of neurons were modulated during social facial touch episodes. On average, across all areas, we found 25.3% touch neurons, and 8.3% sex-touch neurons (Fig. 3f). We found that the proportions of touch and sex-touch neurons depended on the brain area ($p = 0.0096$, $\chi^2$ test of independence, Fig. 2g), and a mosaic plot[20]

**Fig. 3** Single cortical neurons signal social touch and partner sex. **a** Raster plot of example touch (activated S1 L5b neuron and suppressed VMC L5b neuron) and sex-touch neurons (activated VMC L2/3 neuron and suppressed VMC L6 neuron). Raster plots show spike times (black dots) aligned to the first whisker-to-whisker touch in each social-touch episode. Social-touch episodes are sorted by partner sex (female: pink, male: blue) and by duration (indicated by length of colored bar). Many touch episodes happen close together in time, and there is a large variability in the touch duration. **b** Peri-stimulus time histograms of the example neurons shown in **a**, separated by partner sex. Line indicates mean firing rate (smoothing: Alpha kernel, $\tau = 75$ ms), shaded area indicates s.e.m, and pink/blue color indicates female/male partners. **c** Peri-stimulus time histograms of the example neurons shown in (**a**), calculated from the fitted regression model, shown for comparison (plot conventions as in (**b**)). **d** Estimating touch modulation: log-likelihood values of models fitted to the neurons shown in (**a**). The log likelihood of models depending on touch is indicated by the green arrow, the log likelihood of models without touch is indicated by the gray arrow, and the log-likelihood distribution of shuffled touch models is indicated by green bars. All neurons are significant at $p < 0.05$ (the green arrow is outside the shuffled distribution). **e** Log-likelihood values of models fitted to the neurons in (**a**). The log likelihood of models depending on both partner sex and touch is indicated by the brown arrow, the log likelihood of models without sex is indicated by the green arrow, and the log-likelihood distribution of shuffled sex-touch models is indicated by brown bars. The two touch neurons are not significantly modulated by sex (the brown arrows are inside the shuffled distribution); both sex-touch neurons are significant at $p < 0.05$. **f** Number of neurons that are modulated by touch ('touch neurons', green color) and neurons that are modulated by touch, but respond differently to male and female conspecifics ('sex-touch neurons', pink/blue striped color). **g** Mosaic plot of the distribution of touch neurons, sex-touch neurons, and nonsignificant neurons across cortical areas (the $p$ value indicates $\chi^2$ test of independence). Colors indicate significantly increased (dotted) and decreased (gray) proportions (standardized Pearson residuals at $p < 0.05$)

revealed that this was driven by the fact that PrL had less touch neurons and more nonsignificant neurons than other brain areas (all: |standardized Pearson residual| > 1.96, i.e., $p < 0.05$, Fig. 2g). This suggests that although there are differences in the proportions, information about touch and sex of the interaction partner is available across all investigated brain areas.

In a typical experiment, a subject animal would interact with at least two male and two female conspecifics (see Methods). In our analysis so far, we have naively grouped female and male interaction partners together and regressed to find neurons that respond differently depending on the sex of the interaction partner ('sex-touch' neurons). We wondered if this observation was really due to the sex of the partner animal, or if partner-sex dependence could be an artifact of the underlying individual-partner-specific responses. We used an information-theoretical approach[21] and found that even for the neurons most likely to have individual-specific responses, firing rates were significantly more informative about the actual sex than shuffled sex assignments, suggesting that the interaction partner's sex—not individual-partner identity—is the major determinant of touch responses ($p = 0.007$, paired model, Supplementary Fig. 2).

**Population dynamics depend on partner sex and subject sex.** During social facial touch, different cortical regions have different prototypical response patterns[13,17]. By using our regression approach, we found that—as a population—S1 neurons generally increased their firing rate during social touch, that VMC, ACC, and A1 neurons generally decreased their firing rate, and that PrL neurons displayed no modulation at the population level (all assessed at $p < 0.05$, Wilcoxon signed-rank tests, Supplementary Fig. 2a). When we only analyzed neurons where touch modulation was significant at the single-cell level, we found that these generally responded in the same direction as the population as a whole (Supplementary Fig. 2b).

To investigate if the cerebral cortices of male and female rats process social touch differently, we first simply plotted all touch and sex-touch neurons recorded in the respective areas and separated neurons recorded in males and females ('male neurons' and 'female neurons'). In S1, firing rates of both male and female neurons were increased by social touch, and there was no difference between the sexes ($p_{\text{male}}$, $p_{\text{female}} < 0.01$, male vs. female $p > 0.05$, Fig. 4a). In VMC, ACC, and A1, we found that both male and female neurons were generally decreased by touch, and we found no differences between the sexes there either (all: $p_{\text{male}}$, $p_{\text{female}} < 0.05$, except $p_{\text{female}} = 0.065$ for ACC, $p_{\text{male}} = 0.067$ for

A1, all male vs. female $p > 0.05$, Fig. 4a). In PrL, neither male nor female neurons were significantly modulated as a population, and there was no difference between the sexes ($p_{\text{male}}$, $p_{\text{female}} > 0.05$, male vs. female $p > 0.05$, Fig. 4a).

Next, we investigated if population responses depend on the sex of the interaction partner. In the posterior parietal cortex, there is evidence that modulation by sensory stimuli and movement features is distributed randomly across neurons with no or little population structure[22]. To investigate if the same is true during social touch, we plotted the modulation of neurons during interactions with male and female conspecifics. Similar to the posterior parietal cortex, we found that responses were very diverse, and that neurons populated all quadrants (Fig. 4b, Supplementary Fig. 4). Sex-touch neurons were both in the first and third quadrants (corresponding to differences in magnitude of modulation) and second and fourth quadrants (corresponding to different directions of modulation, e.g., increasing during interactions with males and decreasing during interactions with females). As expected from the definition, touch neurons were generally near the diagonal, and nonsignificant neurons were near the origin (Fig. 4b).

First, we focused on the three brain areas where we had a substantial dataset (VMC, S1, and A1). In these areas, we found that responses to male and female interaction partners were highly correlated (VMC/S1/A1: Kendall's $\tau = 0.40/0.38/0.32$, all $p < 0.001$, Fig. 4b). The high correlation values suggest that the population response to male and female stimuli is nonrandom. But what is the structure of the population response, and does it depend on the sex of the subject animal? High correlation values could reflect that there is actually no difference between responses to males and females. This would correspond to having all neurons along the diagonal, and could mean that the sex-touch neurons identified above might simply be neurons at the edge of this unity distribution (Fig. 5a, left). If responses to males and females are different, the responses could be biased toward one sex, responses when touching one sex could be potentiated, or it could be a combination of both. A biased response would, for example, correspond to having all neurons spike more when interacting with males than females (Fig. 5a, middle). A potentiated response would, for example, mean that responses when touching females are always stronger in magnitude than responses when touching males (Fig. 5a, right). Finally, these response patterns might depend on the sex of the subject animal.

To identify the population pattern, we performed mixed-effect generalized linear modeling to investigate how the responses to

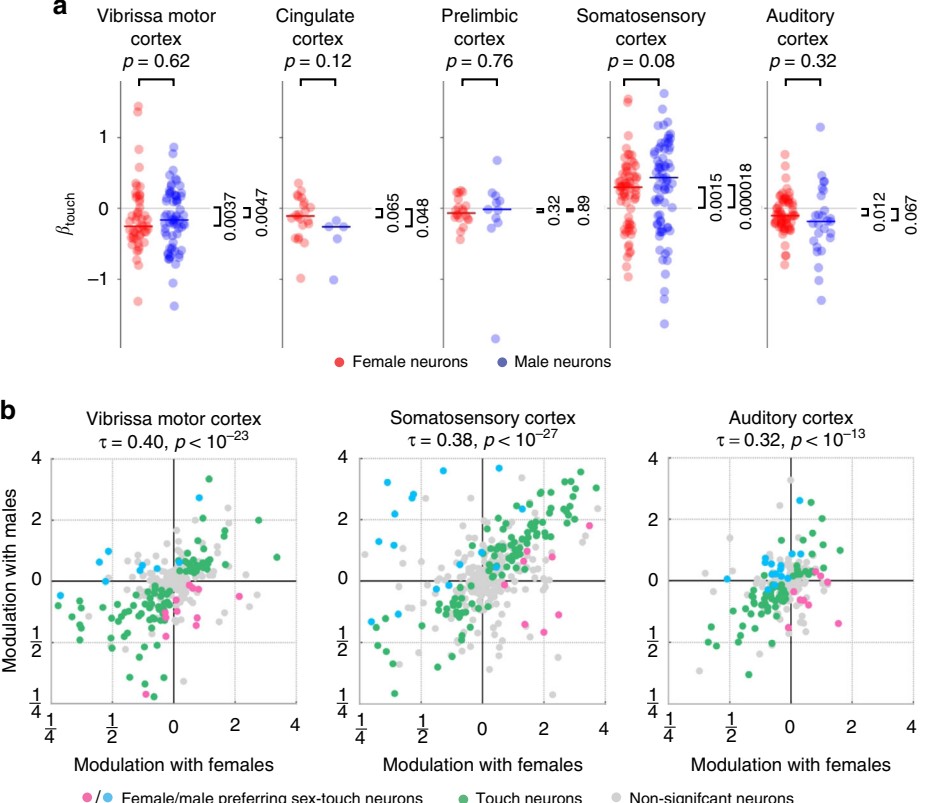

**Fig. 4** Sex-touch responses vary by cortical area and have population structure. **a** Fitted $\beta_{touch}$ of all touch and sex-touch neurons across cortical areas (axis clipped at $+/-1.6$, all data used for calculations, all data plotted in Supplementary Fig. 2a). Colored dots indicate neurons recorded in female (red) and male (blue) animals. In S1, firing rates of both male and female neurons were increased by social touch (median male $\beta_{touch} = 0.44$, $p_{male} = 0.000018$, $t = 4.57$, median female $\beta_{touch} = 0.30$, $p_{female} = 0.0015$, $Z = 3.17$, male vs. female $p = 0.08$, $Z = 1.78$, and $N = 79/71$). In VMC, ACC, and A1, firing rates of both male and female neurons were generally decreased by touch (VMC: median male $\beta_{touch} = -0.16$, $p_{male} = 0.0043$, $t = -2.93$, median female $\beta_{touch} = -0.25$, $p_{female} = 0.0037$, $Z = -2.91$, male vs. female $p = 0.62$, $Z = 0.50$, and $N = 66/51$, ACC: median male $\beta_{touch} = -0.26$, $p_{male} = 0.048$, $t = -2.81$, median female $\beta_{touch} = -0.11$, $p_{female} = 0.065$, $t = -1.95$, male vs. female $p = 0.12$, $t = -1.83$, and $N = 5/22$, and A1: median male $\beta_{touch} = -0.18$, $p_{male} = 0.067$, $t = -1.91$, median female $\beta_{touch} = -0.10$, $p_{female} = 0.012$, $t = -2.58$, male vs. female $p = 0.32$, $t = -0.99$, and $N = 27/67$). In PrL, neither male nor female neurons were significantly modulated as a population, and there was no difference between the sexes (median male $\beta_{touch} = -0.015$, $p_{male} = 0.89$, median female $\beta_{touch} = -0.067$, $p_{female} = 0.32$, $t = -1.02$, male vs. female $p = 0.76$, $Z = 0.31$, and $N = 11/20$). All: $t$ tests ($p_{female}$, $p_{female}$) and unpaired $t$ test with unequal variance (male vs. female) if normal by a Lilliefors test, else Wilcoxon signed-rank tests and Mann–Whitney $U$ test, respectively. **b** Modulation of activity (in fold change) during social touch with male and female conspecifics is highly correlated (VMC/S1/A1: Kendall's $\tau = 0.40/0.38/0.32$, all $p < 10^{-23}/10^{-27}/10^{-13}$). Touch neurons are indicated by green dots, female/male preferring sex-touch neurons are indicated by pink/blue dots, and nonsignificant neurons are indicated by gray dots, Kendall's $\tau$, and $p$ value above

male and female partners depend on the partner-sex-by-subject-sex contingency (see Methods). We did not find any evidence of a bias in responses (all: $p_{intercept} > 0.05$) and no subject-sex-dependent bias (all: $p_{subject\_sex} > 0.05$, full model specification in Supplementary Note 2). However, in all three areas, we found evidence of potentiation. There was a highly significant dependence of responses to males on the response to females (all: $p_{subject\_sex} < 0.001$), and the estimate of the slope was significantly smaller than unity, suggesting that responses to females are stronger in magnitude than responses to males (VMC/S1/A1: $\beta_{subject\_sex}$ [CI] = [0.27–0.49]/[0.54–0.78]/[0.45–0.078], Fig. 5b). In S1 and VMC, there was also a significant interaction between the subject sex and this potentiation (both: $p_{female\_modulation \times subject\_sex} < 0.01$, Supplementary Note 2). Interestingly, the interaction had opposite signs (VMC/S1: $\beta_{female\_modulation \times subject\_sex}$ [CI] = [0.055–0.39]/[–0.54,–0.16], Fig. 5b).

We also found significant correlation values between responses to male and female conspecifics in the areas where we only had a limited sample (ACC/PrL: Kendall's = 0.22/0.17, both $p < 0.01$,

Supplementary Fig. 5a). In these smaller datasets, there was no statistically significant subject-sex-dependent or stimulus-sex-dependent potentiation, but—although not significant—the maximum-likelihood fit also estimated $\beta_{subject\_sex}$ to be less than unity for both ACC and PrL (Supplementary Fig. 5b, Supplementary Note 2). The fact that the pattern is similar in the areas with little data might suggest that this partner-sex-dependent potentiation is very widespread across the cortex. We also note that in ACC we did see a significant bias, indicating that ACC neurons fire less with male than female conspecifics, in both male and female subjects ($p_{intercept} < 0.05$, Supplementary Note 2).

In summary, for both male and female subjects, cortical responses were larger in magnitude with female than male interaction partners (statistically significant in VMC, S1, and A1, same direction, but not significant in ACC or PrL). In S1, the potentiation of responses to females was stronger in female subjects than male subjects; in VMC, the potentiation was stronger in male subjects; in A1, the potentiation did not depend on the subject sex.

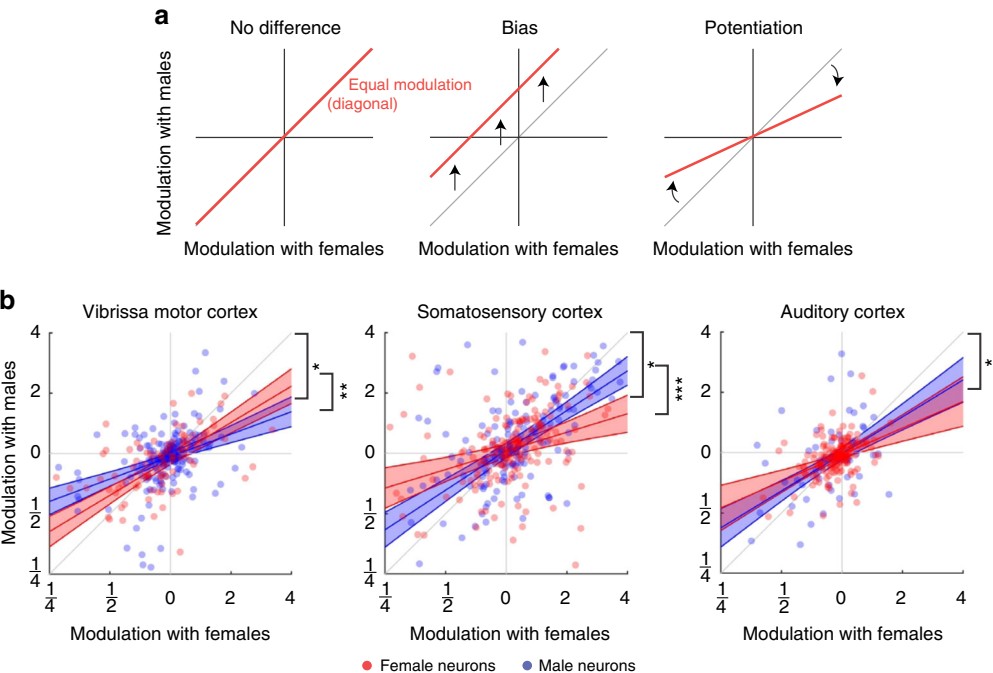

**Fig. 5** Population dynamics depend on both subject sex and partner sex. **a** Some possible types of population structure. If there is no overall difference between responses to male and female conspecifics, neurons will fall on the diagonal (left panel). A biased response to one partner sex corresponds to a shift away from the diagonal. For example, the red line corresponds to a situation where neurons always fire more spikes when touching males than females (middle panel). A potentiated response to one partner sex corresponds to a change in slope of the regression line. For example, the red line corresponds to a situation where neuronal responses to female conspecifics are always larger in magnitude than responses to male conspecifics (right panel). **b** Population response pattern depends on subject sex and partner sex. Dots indicate neurons recorded in female (red) and male (blue) subject animals, lines indicate maximum-likelihood fit of modeling the modulation with males as a function of modulation with females and the sex of the subject animal (red = female subject, blue = male subject, shaded area indicates 95% CI) asterisks (*) indicates slope different from unity (outside 95% CI for both males and females), asterisks (**) indicates $p < 0.01$, asterisks (***) indicates $p < 0.001$ (full model specification in Supplementary Note 2)

**Plausible cellular mechanisms underlying population dynamics.** We found that both partner sex and subject sex were encoded in a potentiation of social-touch responses. This potentiation pattern suggests that a potential mechanism could be a change in inhibitory drive, similar to how inhibitory neuron subtypes control context-dependent modulation of sensory responses in the visual and auditory cortex (for review see ref. [23]). A likely candidate mechanism for the regulation of cortical interneuron activity during social touch is by neuromodulatory hormones, such as estrogen and oxytocin, both key regulators of sociosexual behavior.

While classically thought to mainly act on slow timescales (~hours), estrogen can be rapidly synthesized in the brain and might affect sensory processing at much faster timescales (<min)[24]. We have previously reported that baseline firing rates in putative excitatory neurons in the rat somatosensory cortex are lower during estrus[13], that deep-layer parvalbumin-positive interneurons in the rat somatosensory cortex express estrogen receptor $\beta$, and that fast-spiking interneuron firing rates increase in estrus and with estradiol injection[15]. Despite the change in baseline firing rate, however, we previously found that social-touch responses during staged social interactions in head-fixed animals were unchanged across the estrus cycle, both for regular-spiking and fast-spiking neurons[15]. In our dataset, we have access to the estrus state of female subject animals in a subset of the data (only from somatosensory cortex). In these data, we also found that responses to social touch were unchanged across the estrus cycle (Supplementary Fig. 6).

Oxytocin action on neural circuits plays a major role in cognitive and emotional aspects of sociosexual behavior, such as social bonding, anxiety, and trust, and intranasal oxytocin affects

the subjective pleasantness of identical social-touch stimuli[4]. Oxytocin receptors are expressed in the cortex mainly by interneurons[25,26], and oxytocin acts on interneurons in the auditory cortex to enable pup retrieval by mother mice[27] and modulates interneuron activity in the prefrontal cortex[26] and the olfactory cortex[28] to enable social recognition. Gentle touch and stroking activates oxytocinergic neurons in the paraventricular hypothalamus[29], which project directly to sensory cortices[25]. This provides a potential circuit, by which touch-related oxytocin release can impact cortical interneuron activity to modify cortical responses and network activity patterns during social facial touch. Moreover, since the pattern of cortical oxytocin receptor expression is sex-dependent[25,30,31], this provides a potential basis for our observed sex differences in network activity patterns.

In order to determine how transient oxytocin release might affect cortical population responses during social touch, we simulated the effect of oxytocin release by using a large-scale spiking neural network model of the somatosensory cortex[32,33] (Fig. 6a). This model allows us to simulate the activity of ~1 mm² of the somatosensory cortex (~80.000 modeled neurons, with ~300 million synapses) with realistic synaptic connectivity and projection patterns based on functional and anatomical studies (see Methods). The model has been independently validated multiple times and reproduces known cell-type-specific and layer-specific distributions of firing rates, correlations, and spike delays.

We still know little about how exactly oxytocin modulates neural processing across cortical regions. Interestingly, however, the little that we do know paints a cohesive picture: multiple recent studies investigating oxytocin action in both the neocortex and isocortex (hippocampus) have observed two main effects, one

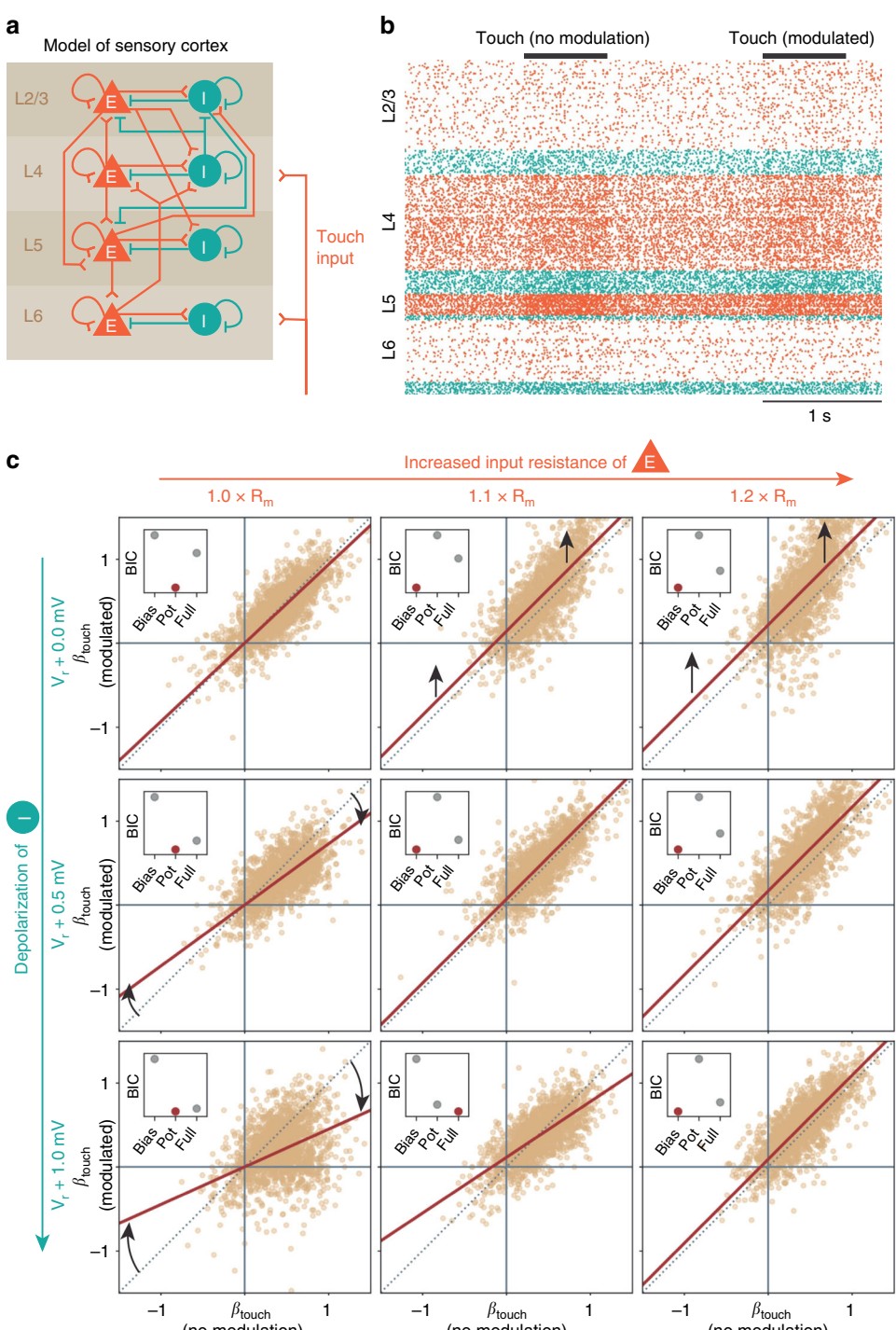

on interneurons and one on excitatory neurons: (i) increased firing rate of inhibitory interneurons, with no change in IPSC amplitude, likely due to membrane depolarization[25–27,34] and (ii) increased excitability of excitatory neurons, visible as an increase in input resistance[35,36].

In our simulations, we thus investigated how a transient depolarization of interneurons and/or a transient increase in the input resistance of excitatory neurons modulated the population response to 'touch' (i.e., thalamic input, two example 'touch' trials shown in Fig. 6b). We made three observations. First, we found that increasing inhibitory drive by depolarizing interneurons leads to a potentiation of the population response, the same type of modulation as we observed during social touch (Fig. 6c, first

column: the slope is different from unity, and there is no shift in the response, compare with Fig. 5b). Second, we found that increasing the input resistance of excitatory neurons simply leads to a bias in responses, not what we observed during social touch (Fig. 6c, first row: there is a constant upward shift, and the slope is not different from unity). Third, we found that when both modulations were simulated simultaneously, the effects "cancel out", and the population response is approximately normalized (Fig. 6c, on the diagonal: the slope is unity with only a very slight bias, intermediate configurations visible in the off-diagonal panels).

As a low-level validation of the modeling approach, we compared the responses of simulated excitatory and inhibitory

**Fig. 6** Modulation of cortical inhibition could underlie population dynamics. **a** We simulated the activity of ~1 mm² of the somatosensory cortex as 77,169 leaky integrate-and-fire neurons with biologically realistic cell-type-specific connectivity (~0.3 billion synapses)[32,33]. The model consists of four cortical layers (2/3, 4, 5, and 6), each with a population of excitatory ("E", orange pyramids) and inhibitory ("I", teal circles) neurons. A population of simulated thalamic neurons provides excitatory synaptic input to layers 4 and 6 to simulate touch. This graphical representation of the model displays the major excitatory and inhibitory projections (connection probabilities > 0.04 are drawn, full connectivity matrix in Methods). **b** Raster plot showing activity of a random subset (2%) of the modeled neurons, during two simulated touch trials (each row is a single neuron, orange/teal dots indicate the spike times of excitatory/inhibitory neurons, and neurons are sorted by layer, simulated touch, indicated by black bars. 'Non-modulated' and 'modulated' trials were interleaved). **c** Population dynamics, when we simulate neuromodulation by oxytocin release during social touch. When we do not modulate the network during touch ($V_r + 0.0$ mV, $1.0 \times R_m$, top left corner), the responses during modulated touch and non-modulated touch are the same (reset voltage of the interneuron populations, $V_r$, and membrane resistance of the excitatory populations, $R_m$, indicated outside plots; each dot indicates the touch response of 2000 single random neurons, the brown line shows the best model, and a small inserted plot indicates the Bayesian information criterion of three models fitted to the data: a 'Bias model' with only a bias and the slope fixed at unity; a 'Potentiation model' with no bias, but a free-slope parameter; a 'Full model' allowing for both a bias and a potentiation of responses, cf. Fig. 5a). The simulations invite three conclusions: increasing the inhibitory drive by depolarizing interneurons leads to a potentiation of touch responses, just as we observed in our data (first column: slope different from unity, no shift, cf. Fig. 5b). Increasing the input resistance of excitatory neurons simply leads to a bias in responses (first row: upward shift from unity line). When both effects are applied simultaneously, the effects "cancel out", and the touch response is approximately normalized (diagonal: the best model again essentially falls on the unity line)

neurons with the response of putatively inhibitory and excitatory neurons in our somatosensory cortex data (assessed by spike shape, see Methods). In real data from S1 (Supplementary Fig. 7b–d), as well as in data from A1 and VMC (Supplementary Fig. 7e–j), we found that responses of putatively inhibitory and excitatory neurons were similar, and in the same direction. This overlap in response magnitudes, and the similar direction of firing rate modulation, agrees with the prediction of the model (Supplementary Fig. 7a).

From these simulations, we draw two conclusions. First, we conclude that changes in inhibitory drive—for example mediated by oxytocin release—are a plausible candidate mechanism, which could explain the population response potentiation during social facial touch. Second, as an additional incidental finding, we reason that the increased excitability of excitatory neurons that is observed after sustained bath application of oxytocin[35,36] might be a homeostatic response attempting to re-normalize network processing in reply to oxytocin-mediated increases in inhibition during such in vitro experiments.

## Discussion

A previous study found that in posterior parietal cortex, firing rate modulation by sensory stimuli and movement features was distributed randomly across neurons with no or little population structure[22]. In contrast to this observation, we found that social-touch responses were highly structured, and that the network activity pattern signaled the sex of the interaction partner by partner-sex-dependent potentiation of responses. Moreover, in the somatosensory and vibrissa motor cortex, we found that the sex of the subject animal itself was encoded by a subject-sex-dependent magnitude of this potentiation. In line with the encoding of partner sex observed, there is previous evidence of population coding of touch stimuli in the somatosensory cortex[37]. Previous investigations in the rodent whisker system have found that putatively 'simpler' stimuli presented in highly controlled conditions, such as object location[38] and texture roughness[39], are encoded with higher fidelity by precise spike timing than by firing rates. However, in line with our observations, previous investigations also found that firing rates carry extra information in addition to the information conveyed by spike timing[40]. The relevant timescale for temporal coding by spike timing in the somatosensory cortex seems to be within tens of milliseconds from the stimulus, and the analysis requires extremely precise information about stimulus-onset time, which is not available in our naturalistic experimental paradigm. More generally, structured population dynamics in vibrissa motor cortex aligns well

with other observations of population coding in the motor cortex, where the population activity vector of both preparatory and movement-related activity correlates with movement features[41].

In line with our observations of touch and sex-touch neurons in the rat cingulate cortex, previous human studies have found that social context strongly modulates responses in the cingulate and orbitofrontal cortex[3–5]. We found that—when examined at single-cell resolution—the majority of cingulate neurons decrease in activity during social touch (likely a positive social interaction). This is in stark contrast to a recent report that only ~1% of rat cingulate-recording sites show decreases in multiunit activity during a painful stimulus, during playback of fear-conditioned tones, or when observing conspecifics receive painful electric shocks[42]. In light of the tight anatomical integration of the pre-limbic cortex with brain structures responsible for social behavior and emotions[43], and the putative homology with human and primate prefrontal cortex, a key structure in social cognition[44], we found it curious that the prelimbic cortex was so weakly modulated by social touch. We did not see overall changes in the firing rate, only weakly modulated touch and sex-touch neurons at both edges of a zero-centered distribution. While previous rodent studies did not investigate social touch as such, activity of the subpopulation of prelimbic neurons is modulated by the presence of conspecifics[45], but—in line with our observations—it has been shown that during social approach (to a caged conspecific), prelimbic responses are generally weak and diverse[46,47], and that social behaviors are encoded by sparse groups of neurons either increasing or decreasing in activity during behavior ('on' and 'off' ensembles[48]).

In line with what we described for other whisking behaviors[17], we found that despite active whisker movement, neurons in vibrissa motor cortex mainly decrease their activity during social touch. We also found a large proportion of sex-touch neurons in vibrissa motor cortex. We already know that neurons in frontal rodent motor cortex can have low-latency responses to touch, and that the motor cortex is an important node in a distributed network for touch processing and sensorimotor cognition (for reviews, see refs. [49,50]). Our study adds a potential role in sex-dependent processing of social-touch cues to this complex picture.

In the somatosensory and motor cortex, we found that social touch leads to different network activity patterns in male and female subjects. Identifying such potential differences in how male and females cortices process social stimuli could help shed light on the biological basis of sex differences in the etiology, prevalence and symptoms of autism, and depression and anxiety

disorders, all characterized by social dysfunction[51–53]. Generally, while there clearly are some systematic sex differences in both anatomy[54–59] and functional connectivity[60], male and female cortices are overall remarkably similar. For example, even though male and female genital anatomy is extremely different, the layout of the somatosensory body map is essentially identical in both sexes[61,62] and has similar projection targets[63]. In recent years, several studies have identified sex differences in subcortical circuits involved in regulating social behavior. For example, circuits involved in the control of aggression in the ventromedial hypothalamus are wired differently in males and females[64], medial amygdala has striking sex differences in olfactory responses[65], and galanin-positive neurons in the medial preoptic area[66] and their subcortical input nuclei[67] are activated during parental behavior in a sex- and reproductive-state-specific manner. In contrast to these subcortical examples, sex differences in cortical processing are rare and subtle[68,69].

Our data invite two, not mutually exclusive, interpretations. On one hand, the differences in processing may be a signature of a universal sex difference in how male and female cortices process the same sensory stimuli. Thus, these sex differences might generalize across other cortical regions and sensorimotor modalities. On the other hand, male and female cortices may use identical computational strategies, and our observed differences simply reflect the fact that for male and female subjects, the same partner sex presents a different social situation with different cognitive and behavioral challenges. For example, a male rat interacting with another male rat might be mainly assessing dominance and aggression, while a female rat meeting a male rat might be assessing aggression as well as reproduction.

We still know very little about sex differences in human cortical processing of social-touch stimuli. As noted in the introduction, our investigation of partner-sex-dependent differences in touch responses was based on pioneering studies with human subjects[3–5]. These studies have not reported sex differences in the processing of social-touch stimuli, but then again, previous studies have not compared experimental subjects of both sexes[3–9,70–72]. It would be most interesting to know if our observed network activity differences in the rat are paralleled in primates and humans. Our findings highlight the importance of complementing human studies with the single-cell resolution offered by animal studies.

During social facial interactions, the actual haptic input from male and female partners is very similar[11]. However, our naturalistic paradigm is inherently multisensory, and even though the whisking and touch input is similar, male and female partners convey very dissimilar olfactory cues. The vomeronasal organ is important for determining the sex of the conspecific in both male–male and male–female interactions, and might be an important 'bottom-up' pathway for the modulation of cortical responses during social touch observed here[12]. Still, other sensory modalities might also contribute. For example, there might be subtle sex differences in whisking patterns[11] or vocalizations[73]. Our definition of social 'touch' neurons is purely descriptive, and we do not know what low-level features of the social-touch episode (haptic touch, ultrasonic calls, olfactory cues, and temperature) or internal top–down processes modulate the firing rate of these neurons.

We found that changes in social context (partner sex and subject sex) were associated with a structured potentiation of social-touch responses. The neuromodulatory hormone oxytocin is a prime candidate for cortical neuromodulation during social interactions[74], and our simulations suggest that changes in inhibitory activity by social-touch-dependent release of oxytocin could plausibly underlie our observed context-dependent modulation of touch responses. Across all areas, we observed that responses to female partners tended to be bigger in magnitude than responses to male partners. Thus, if the touch response potentiation is indeed mediated by oxytocin via the simple mechanisms outlined in our simulations, this makes the prediction that oxytocin release should be larger when interacting with a male than a female partner. Does this prediction match what has been reported in the literature? Surprisingly, there is a real lack of comparative studies between males and females[31,74], and we could only find two reports that speak to this question. When heterosexual women touch their male partners, they release more systemic oxytocin than heterosexual men do when touching their female partners[75], and male rats release more systemic oxytocin when socially interacting with male partners than female rats do when socially interacting with female partners[76]. These two observations are obviously consistent with our prediction, but they constitute extremely weak and indirect evidence. We still know very little about what drives cortical oxytocin release on a moment-to-moment basis during social interactions, and whether cortical oxytocin release depends on the sex of the interaction partner[74].

In any case, it is important to stress that neuromodulation by oxytocin is just one among many possible mechanisms. Meeting a whisking, vocalizing, and pheromone-scented conspecific en face is presumably a highly salient event, and the response modulation across cortical networks could reflect a generalized change in cortical state (alertness, elicited by the high-stimulus salience), perhaps to increase the signal to noise of sensory processing[77]. The social-context-dependent changes in touch responses observed during social facial touch might be partly or fully mediated by other neuromodulators, such as vasopressin, another peptide hormone involved in the control of social behavior[31]; dopamine, which might strongly modulate cortical activity during social motivation and reward[78]; acetylcholine or noradrenaline, both known to modulate cortical state and cortical sensory processing in a context-dependent manner[23].

## Methods

**Animal welfare.** All experimental procedures were performed according to German animal welfare laws under the supervision of local ethics committees. Rats (Wistar) were purchased from Janvier Labs (Le Genest-Saint-Isle, France). Rats presented as partner animals were housed socially in same-sex cages, and post-surgery implanted animals were housed in single animal cages. All animals were kept on a 12:12-h reversed light/dark cycle, and all experiments were performed in the animals' dark phase. Rats had ad libitum access to food and water.

**Social gap paradigm.** Behavioral experiments were done by using the social gap paradigm[11] (Fig. 1a). The experimental paradigm consists of two elevated platforms, 30-cm long and 25-cm wide, surrounded by walls on 3 sides, positioned approximately 20 cm apart. The distance between platforms was varied slightly depending on the size of the rats. The platforms and platform walls were covered with soft black foam mats to provide a dark and non-reflective background, and to reduce mechanical artifacts in tetrode recordings. All experiments were performed in darkness or in dim light, and behavior was recorded from above under infrared light. The implanted rat was placed on one platform, and on the other platform, we either presented various objects or other rats. The implanted rats were not trained, just habituated to the setup and room, and spontaneously engaged in social facial interactions. The rat behavior was recorded at low video speed from above with a 25- or 30-Hz digital camera, synchronized to the electrophysiological data acquisition by using TTL pulses. Typically, recording sessions were performed in four to eight 10–15-min blocks, where we would present either one or two conspecifics (of both sexes) in each block, randomly, see refs. [13,16,17]. Video frames were labeled blind to the spike data.

**Tetrode recordings and histology.** In tetrode-recording experiments, we used ~p60 Wistar rats, which were handled for 2–3 days before being implanted with a tetrode microdrive. Surgery was done as previously described[13]. The implanted microdrive had eight separately movable tetrodes driven by screw microdrives (Harlan 8-drive; Neuralynx, Bozeman, MT, USA). The tetrodes were twisted from a 12.5-μm-diameter nichrome wire coated with polyimide (California Fine Wire Company), cut and examined for quality by using light microscopy, and gold-plated to a resistance of ca. 300 k Ohm in the gold-plating solution by using an automatic plating protocol ("nanoZ", Neuralynx). For tetrode recordings targeting

VMC, ACC, and PrL, a craniotomy of $1 \times 2$ mm was made 0.75–2.75 mm anterior and 1–2 mm lateral to bregma[79,80]. ACC includes data from the area 'Cg1' and dorsal 'Cg2' in the rat atlas[79]. A1 includes data recorded in the rat atlas regions 'Au1','AuD', and 'AuV'[79]. Steel screws for stability and two gold-plated screws for grounding the headstage were drilled and inserted into the skull, and the gold-plated screws were soldered and connected to the headstage PCB by using silver wire. After fixation of all screws, the dura was removed, the implant fixated above the craniotomy, the craniotomy sealed with 0.5% agarose, and the tetrode drive fixed in place with dental cement (Heraeus). The tetrodes were arranged in a 2-by-4 grid with a spacing of ~500 μm. Neural signals were recorded through a unity-gain headstage preamplifier and transmitted via a soft tether cable to a digital amplifier and A/D converter (Digital Lynx SX; Neuralynx) at 32 kHz. We filtered the signal between 600 and 6 kHz and detected spikes by crossing of a threshold (typically ~50 μV) and saved each spike (23 samples, 250 μs before voltage peak to 750 μs after voltage peak). At the end of the experiment, animals were again anesthetized with a mix of ketamine and xylazine, and the single-tetrode tracks were labeled by using small electrolytic lesions made by injecting current through the tetrode wire (10 μA for 10 s, tip-negative DC). After lesioning, animals were perfused with phosphate buffer followed by a 4% paraformaldehyde solution (PFA). Brains were stored overnight in 4% PFA before preparing 150-μm coronal sections. Sections were stained for cytochrome oxidase to reveal the areal and laminar location of tetrode- recording sites, which could be calculated from the location of tetrode tracks and lesions. We only analyzed data from recording sites where the lesion pattern could unanimously identify the tetrode and the recording sites.

All spike analysis was done in Matlab (MathWorks, Natick, MA, USA). Spikes were preclustered off-line on the basis of their amplitude and principal components by means of a semiautomatic clustering algorithm ('KlustaKwik', https://github.com/klusta-team/klustakwik[81]). After preclustering, the cluster quality was assessed and the clustering refined manually by using MClust (http://redishlab.neuroscience.umn.edu/MClust/MClust.html, A. D. Redish, University of Minnesota). The spike features used for clustering were energy and the first principal component of the waveform. To be included in the analysis as a single unit, clusters had to fulfill the following criteria: first, the L ratio, a measure of distance between clusters, was below 0.5. s, the histogram of inter-spike intervals (ISIs) had to have a shape indicating the presence of single units, e.g., a refractory time of 1–2 ms, or the appearance of a bursty cell (many short ISIs). Multiunit clusters were not included in the analysis.

**Previous use of data**. Part of the data presented in this study has already been presented in other studies. The barrel cortex data, recorded throughout all cortical layers, have previously been published in ref. [13]. The auditory cortex data, recorded throughout all cortical layers, have previously been published in ref. [16]. The recordings from deep layers of vibrissa motor cortex have previously been presented in ref. [17]. The previous study on barrel cortex[13] investigated how touch-evoked activity depended on male and female subject and partner animals in a PSTH-based way (like our Fig. 1c, d); the other two previous studies did not investigate sex differences[16,17]. Neuons were recorded throughout the cortical column, with a majority of neurons in the deep layers. The laminar distribution was (S1/VMC/PrL/ACC/A1): layer 6: 32/16/39/31/46, layer 5b: 88/153/70/17/100, layer 5a: 42/37/33/47/63, layer 4: 56/0/0/0/19, layer 2/3: 37/90/0/0/4, and layer uncertain: 129/0/0/0/7. All data from the superficial layers of vibrissa motor cortex, data from all layers of the cingulate cortex, and data from all layers of the prelimbic cortex have not previously been published.

**Peri-stimulus time histograms**. To identify significantly increasing and decreasing neurons, as shown in Fig. 1c, d, we calculated the mean firing rate in the 'baseline' period (−2500 to 0 ms) and the poststimulus period (0–500 ms) and determined significant changes by using Wilcoxon signed-rank tests. For display, the PSTHs were smoothed with PSP-shaped alpha function (i.e., $f(x) = \frac{x}{\tau} \exp(1 - \frac{x}{\tau})$) with $\tau = 75$ ms.

**Statistical modeling to identify touch and sex-touch neurons**. We used a generalized linear regression approach[18,19] to identify touch and sex-touch responses. We discretize the spike train in 1-ms bins, reduced the data amount by removing baseline periods, which were more than 5 s from any social interaction, and model the firing rate as a Poisson process. If we assume that the discharge of spikes within each time bin is generated by a homogeneous Poisson point process, then the probability of observing $y$ spikes in a single time bin is

$$p(y|\lambda) = \frac{(\lambda\Delta)^y}{y!} \exp(-\lambda\Delta) \quad (1)$$

where $\Delta = 1$ ms is the width of the time bin and $\lambda > 0\,\mathrm{s}^{-1}$ is the expected discharge rate of the cell. If we assume that each time bin is independent, the probability of the entire spike train, $\bar{y}$ is

$$p(\bar{y}|\bar{\lambda}) = \prod_i \frac{(\lambda_i\Delta)^{y_i}}{y_i!} \exp(-\lambda_i\Delta) \quad (2)$$

where $y_i, \lambda_i$ is the observed number of spikes and the expected discharge rate in the

$i$'th time bin, respectively. If we model the expected discharge rate, $\bar{\lambda}$, so that it depends on the parameters, $\bar{\beta}$, we have the log-likelihood function

$$\log\mathcal{L}(\bar{\beta}) = \log p(\bar{y}|\bar{\lambda}(\bar{\beta})) = \sum_i y_i \log\lambda_i + \sum_i y_i \log\Delta - \sum_i \log y_i! - \Delta\sum_i \lambda_i$$

$$(3)$$

We model $\bar{\lambda}$ so that it depends linearly on spike history, experimental recording, touch, and partner sex, and—since the expected firing rate cannot be negative—we model

$$\bar{\lambda} = \exp(\mathbf{P} \cdot \bar{\beta}) \quad (4)$$

where $\mathbf{P}$ is a predictor matrix and $\bar{\beta}$ is a vector of regression coefficients[18]. The predictor matrix has the following columns: a constant baseline rate; five 1-ms spike history bins; six 25-ms history bins; $(N_{rec}-1)$ one-hot columns to model a change in baseline between the recordings; a one-hot column indicating all social-touch episodes; a one column indicating the sex of the stimulus animal (0 = female, 1 = male). Due to the refractory period of the cell, it is not correct to assume that all time bins are statistically independent, so following ref. [19], we include 11 spike history parameters, $h_1...h_{11}$, to model the inter-spike-interval distribution of the cell. The spike history is binned up into 11 successive bins, five 1-ms bins (vectors with no. of spikes in the previous 0–1 ms, 1–2 ms, 2–3 ms, 3–4 ms, and 4–5 ms) and six 25-ms bins (vectors with no. of spikes in the previous 5–30 ms, 30–55 ms, 55–80 ms, 80–105 ms, 105–130 ms, and 130–155 ms). We also include constant bias terms to allow for variations in baseline firing rate between each recording. Thus,

$$\bar{\beta} = \left[\beta_0, h_1 ... h_{11}, \beta_{rec_1} ... \beta_{rec_{(n-1)}}, \beta_{touch}, \beta_{sex}\right] \quad (5)$$

and in Wilkinson notation, the linear model would be written as

$$y \sim \beta_0 + \sum_i^{11} h_i + \sum_i^{n-1} \beta_{rec_i} + \beta_{touch} + \beta_{sex} \quad (6)$$

An example predictor matrix is shown in Fig. 2. We used the function package 'neuroGLM' (https://github.com/pillowlab/neuroGLM, ref. [82]) to calculate and numerically fit the models in Matlab (MathWorks, Natick, MA, USA). To assess the statistical significance of touch and sex-touch responses, we used a nonparametric, shuffling-based model selection approach. To assign statistical significance to $\beta_{sex}$, we compared the log likelihood of the model including touch and sex as predictors with the distribution of log-likelihood values from the same models fitted to predictor matrices, where we randomly shuffled the label (male/female) of the partner animal ($N = 100$, Fig. 3d). We do not assume that random effects have a Gaussian distribution. Rather, to assign statistical significance to $\beta_{touch}$, we compared the log likelihood of the model including touch (and not sex) as predictors with the distribution of log-likelihood values from the same models fitted to predictor matrices, where we circularly permutated the column indicating where the social-touch episodes had happened ($N = 100$, Fig. 3e). We defined 'sex-touch neurons' as neurons with $p_{sex} < 0.05$, 'touch neurons' as neurons with $p_{sex} > 0.05$ and $p_{touch} < 0.05$, and 'non-significant' neurons as neurons with both $p_{sex} > 0.05$ and $p_{touch} > 0.05$. We compared the proportions of nonsignificant, touch, and sex-touch neurons across cortical areas by calculating standardized Pearson residuals and visualized the contingency table as a mosaic plot. Standardized residuals and mosaic plots were generated by using the 'vcd' package[20] for R.

**Information theory**. To avoid assuming anything about the direction of firing rate changes, we calculated the information per spike

$$I_{pr.spike} = \frac{1}{\lambda} \sum_i \lambda_i \log_2 \frac{\lambda_i}{\lambda} p_i \quad (7)$$

where $\lambda$ is the average spike rate, $\lambda_i$ is the spike rate during the $i$'th stimulus, and $p_i$ is the probability of the $i$'th stimulus[21]. To identify neurons, which could potentially have individual-specific or sex-specific response patterns, we treated as 'stimuli' the social-touch episodes with all individual-partner animals. Significance was assessed by a shuffling procedure, where we circularly shifted the timing of the social-touch episodes over the recordings ($N = 200$, $p < 0.05$). To investigate if these putatively individual-specific neurons carry more information about the real sex of the partner animal than randomly assigned sex labels, we also used a shuffling procedure. For all neurons, we calculated the mutual information per spike with three situations, baseline, social touch with male partners, and social touch with female partners, and calculated the same value, where we shuffled the sex of the partner animals (Supplementary Fig. 3a). Since the number of possible shuffles depend on the number of partner animals, the dataset had strong dependency per neuron[83], so we fitted a mixed-effects model

$$\Delta I \sim 1 + (1|neuron) \quad (8)$$

where $\Delta I$ is the real minus the shuffled value of the information per spike between the three 'stimuli'.

**Calculating the magnitude of responses**. To estimate the average depth of modulation of increasing and decreasing neurons across both partner sexes (Fig. 3),

we calculated the fold modulation by touch as the ratio

$$\text{Touch mod.} = \frac{\lambda_{\text{touch}}}{\lambda_{\text{baseline}}} = \frac{\exp(\beta_0 + \beta_{\text{touch}})}{\exp(\beta_0)} = \exp(\beta_{\text{touch}}) \quad (9)$$

where $\lambda_{\text{touch}}, \lambda_{\text{baseline}}$ are the firing rates and $\beta_{\text{touch}}$ is the fitted regression coefficient of the regression models including touch (and not sex) as a predictor. To plot and compare the magnitude of increases and decreases, we calculated the numerical value of the base-2 logarithm and plotted the data as fold increases/decreases (e.g., $\log_2$ (ratio) = 1 corresponds to a twofold increase (double the firing rate), $\log_2$ (ratio) = –1 corresponds to a twofold decrease (half the firing rate), etc.). To estimate the modulation when touching male and female conspecifics (Fig. 4 and 5), we calculated the fold modulation from the fitted models as

$$\text{Female mod.} = \frac{\lambda_{\text{female touch}}}{\lambda_{\text{baseline}}} = \frac{\exp(\beta_0 + \beta_{\text{touch}})}{\exp(\beta_0)} = \exp(\beta_{\text{touch}}) \quad (10)$$

$$\text{Male mod.} = \frac{\lambda_{\text{male touch}}}{\lambda_{\text{baseline}}} = \frac{\exp(\beta_0 + \beta_{\text{touch}} + \beta_{\text{sex}})}{\exp(\beta_0)} = \exp(\beta_{\text{touch}} + \beta_{\text{sex}}) \quad (11)$$

where $\lambda_{\text{female touch}}, \lambda_{\text{male touch}}, \lambda_{\text{baseline}}$ are the firing rates and $\beta_{\text{touch}}, \beta_{\text{sex}}$ are the fitted regression coefficients of the full GLM models including touch and sex (i.e., this is a different fitted value of $\beta_{\text{touch}}$ than above, Touch mod.≠Female mod.). To determine the population response pattern of the modulation, we fitted the mixed-effects regression

$$\text{male\_mod} \sim 1 + \text{female\_mod} + \text{subject\_sex} + \text{female\_mod} \\ \times \text{subject\_sex} + (1|\text{subject}) \quad (12)$$

where male_mod and female_mod are the modulation ratios, subject_sex is a one-hot vector indicating the sex of the subject animal (0 = female, 1 = male), and subject is a categorical variable indicating the subject animal. To avoid biasing the regression fit by the few cells with extremely low-firing rates (indicated by circles in Supplementary Fig. 4a), we removed neurons with a more than 32-fold increase/decrease from the fit. The (1|subject) is a constant error term per subject animal, which we add due to the dependence introduced by the fact that we have unequal numbers of neurons recorded from the different subject animals[83].

**Spiking neural network model.** We simulated the activity patterns of ~1 mm$^2$ of the sensory cortex as 77,169 leaky integrate-and-fire (LIF) neurons with biologically realistic cell-type-specific connectivity. The model was developed by ref. [32], and simulated spontaneous activity patterns, transmission delays, and cell-type-specific firing rates are in agreement with in vivo recordings in awake animals[32]. In our simulations, we modified the implementation of the model by ref. [33], and ran our simulations in the 'Brian 2' spiking neural network simulator for python[84]. The simulated neurons are distributed over eight populations, corresponding to excitatory and inhibitory neurons in layers 2/3, 4, 5, and 6. In addition, a population of thalamic neurons provide 'touch' input. The numbers of modeled neurons are given in Methods Table 1.

The neural populations are connected with excitatory and inhibitory synapses. For every possible pre- to postsynaptic population, the model assigns $K$ synapses

$$K = \frac{\log(1 - p_{\text{conn}})}{\log(1 - 1/(N_{\text{pre}} \cdot N_{\text{post}}))} \quad (13)$$

where $N_{\text{pre}}$ and $N_{\text{post}}$ are the total number of neurons in the populations (given in Methods Table 1) and $p_{\text{conn}}$ is the connection probability (given in Methods Table 2). Once the total number of synapses is calculated, all the synapses are randomly assigned between neurons in the pre- and postsynaptic populations (pairs of neurons are drawn from a uniform distribution with replacement and $K \gg N$, so there are multiple synapses per neuron).

The membrane potential, $V(t)$, of an LIF neuron is governed by the differential equation

$$\frac{dV}{dt} = -\frac{V(t) - V_r}{\tau_m} + \frac{I_{\text{ext}}(t) + I_{\text{syn}}(t)}{C_m} \quad (14)$$

where $\tau_m = 10$ ms is the membrane time constant, $C_m = 250$ pF is the membrane capacitance, $V_r = -65$ mV is the reset potential, $I_{\text{ext}}(t)$ is a background current, and $I_{\text{syn}}(t)$ is the synaptic current. When the membrane potential of a neuron reaches a threshold value, $V_{\text{th}} = -50$ mV, the neuron spikes and is reset and held at a reset voltage, $V_r = -65$ mV, for a refractory period of $\tau_{\text{ref}} = 2$ ms. When a presynaptic neuron spikes, the postsynaptic neuron receives an increase in synaptic current of $w_e$

(the synaptic weight) for excitatory neurons and $w_i = -4w_e$ for inhibitory neurons. $w_e$ is drawn from a normal distribution, Normal ($\mu = 87.8$ pA, $\sigma = 8.8$ pA). The synaptic weight for connections from neurons in L4e–L23e is doubled. Synaptic currents are delivered with a delay $d_e = \text{Normal}(\mu = 1.5 \text{ ms}, \sigma = 0.75 \text{ ms})$ for excitatory synapses and $d_i = \text{Normal}(\mu = 0.8 \text{ ms}, \sigma = 0.4 \text{ ms})$ for inhibitory synapses. Synaptic current changes are modeled as exponentials with a time constant $\tau_{\text{syn}} = 0.5$ ms and governed by the differential equation

$$\frac{dI_{\text{syn}}}{dt} = -\frac{I_{\text{syn}}(t)}{\tau_{\text{syn}}} \quad (15)$$

We initialized the simulation with an initial membrane potential drawn from Normal ($\mu = -58$ mV, $\sigma = 10$ mV). The network is driven by delivering a layer-specific background current to all neurons, $I_{\text{ext}}(t) = b \cdot 0.3512$ pA, where $b$ is a layer-specific background-scaling factor. The background-scaling factor, $b$, is given in Methods Table 3.

At baseline, the network does not receive input from the thalamic population (firing rate of thalamic neurons, $r_{\text{thal}} = 0$ Hz). Each simulated touch trial started with a baseline period of 1000 ms, then we simulated 'touch' by increasing $r_{\text{thal}}$ to 30 Hz for a period of 700 ms, and then we again simulated a baseline period of 300 ms (see Fig. 6b for two example trials). In every other trial, we simulated a change in the network, stemming from social-touch-mediated release of oxytocin during touch. In order to simulate a depolarization-induced increase in the firing rate of interneurons during touch, we increased the reset potential, $V_r$, of the interneuron populations by [0 mV, + 0.5 mV, and + 1.0 mV]. In order to simulate an increase in input resistance of excitatory neurons, we multiplied the membrane time constant, $\tau_m$, with [1.0, 1.1, and 1.2], only during touch. We scale the membrane time constant $\tau_m$, since $\tau_m = R_m \cdot C_m$, where $R_m$ is the membrane resistance and $C_m$ is the membrane capacitance. All simulations were done by using the 'Brian 2' exact integration method for linear equations, with 0.1-ms time steps. For every configuration, we performed 20 non-modulated and 20 modulated trials (interleaved), and calculated $\beta_{\text{touch}} = \log(r_{\text{touch}}/r_{\text{base}})$ for 2000 neurons chosen randomly across all populations. We estimated the rates $r_{\text{base}}$ and $r_{\text{touch}}$ by counting spikes in a 700-ms window before and after the simulated-touch onset, respectively. We removed extremely low-firing neurons (neurons with less than three spikes across trials). In order to characterize the network response pattern, we fit three models to the modulated and non-modulated responses: a 'Bias model' with only a bias and the slope fixed at unity; a 'Potentiation model' with no bias, but a free-slope parameter; a 'Full model' with both a bias and a potentiation of responses:

$$\text{Bias model}: Y \sim 1, \quad \text{where } Y = \beta_{\text{modulated}} - \beta_{\text{non-modulated}} \quad (16)$$

$$\text{Potentiation model}: Y \sim X, \quad \text{where } X = \beta_{\text{non-modulated}} \text{ and } Y = \beta_{\text{modulated}} \quad (17)$$

$$\text{Full model}: Y \sim 1 + X, \quad \text{where } X = \beta_{\text{non-modulated}} \text{ and } Y = \beta_{\text{modulated}} \quad (18)$$

The models were fit by using the 'statsmodels' python package. For all three models, we calculated the Bayesian information criterion and used this to select the model, which best characterizes the data (indicated by a red dot in the small inserted plots in Fig. 6c).

**Separation by extracellular spike shape.** As a low-level check of our network simulations, we compared the touch responses of putatively excitatory and inhibitory neurons in the somatosensory cortex (based on their extracellular spike shape) with the response of simulated excitatory and inhibitory neurons in the model. In some brain regions, for example, the hippocampus, separation of cell type based on extracellular spike shape, appears to work well and to reliably identify thin-spiked neurons that suppress simultaneously recorded neurons with short latency[85]. However, in other brain regions the use is more controversial. For example, motor cortical pyramidal projection neurons have extremely thin spikes, and some cortical interneurons have very wide spikes[86]. Further, since the spike width and shape depend strongly on the relative location of the electrode and the cell[87], spike shapes recorded with tetrodes in the agranular rat frontal cortex (with the lack of cytoarchitectonic stereotypy) are likely less reliable than spike shapes recorded in the hippocampus. Following our previous report on S1[13], we used two features of the spike shape: the spike width (at half-maximum) and the 'post-positivity' (the integral of the spike waveform between 0.375 ms and 0.75 ms after the spike peak, normalized by peak voltage). These are slightly different features from our previous report, focusing on A1[16]. For feature calculations, we over-sampled the average spike at twice the sampling rate with cubic-spline interpolation. For clustering, we standardized both features (by z scoring) and fitted a Gaussian mixture distribution with two components ('fitgmdist' in Matlab, with a regularization of 0.1), and clustered the data based on the fitted components. For S1 (Supplementary Fig. 7b, c) and A1 (Supplementary Fig. 7e–g), this approach was feasible: the 2d distribution of features was bimodal, and we could fit a two-component Gaussian mixture model. As we have previously reported[17]—and in line with the motor-cortex-specific caveats outlined above—we did not see a bimodal 2d distribution of features in the vibrissa motor-cortex data, and we could not fit two well-separated Gaussian components. Thus, here we simply fitted a single 2d Gaussian, and 'cut' the 2d distribution through the mean, along the shortest axis (Supplementary Fig. 7h–j). We excluded seven neurons from this analysis because the spike shapes were either not properly stored in our database

**Table 1 Number of modeled neurons**

**Number of modeled neurons**

| L2/3e | L2/3i | L4e | L4i | L5e | L5i | L6e | L6i | Thal. |
|---|---|---|---|---|---|---|---|---|
| 20,683 | 5834 | 21,915 | 5479 | 4850 | 1065 | 14,395 | 2948 | 902 |

## Table 2 Connection probabilities

| | | Connection probability ($p_{conn}$) from... | | | | | | | | |
|---|---|---|---|---|---|---|---|---|---|---|
| | | L2/3e | L2/3i | L4e | L4i | L5e | L5i | L6e | L6i | Thal. |
| Target | L2/3e | 0.101 | 0.169 | 0.044 | 0.082 | 0.032 | 0.0 | 0.008 | 0.0 | 0.0 |
| | L2/3i | 0.135 | 0.137 | 0.032 | 0.052 | 0.3075 | 0.0 | 0.004 | 0.0 | 0.0 |
| | L4e | 0.008 | 0.006 | 0.05 | 0.135 | 0.007 | 0.0003 | 0.045 | 0.0 | 0.0983 |
| | L4i | 0.069 | 0.003 | 0.079 | 0.160 | 0.003 | 0.0 | 0.106 | 0.0 | 0.0619 |
| | L5e | 0.100 | 0.062 | 0.051 | 0.006 | 0.083 | 0.373 | 0.020 | 0.0 | 0.0 |
| | L5i | 0.055 | 0.027 | 0.026 | 0.002 | 0.060 | 0.316 | 0.009 | 0.0 | 0.0 |
| | L6e | 0.016 | 0.007 | 0.021 | 0.017 | 0.057 | 0.020 | 0.040 | 0.225 | 0.0512 |
| | L6i | 0.036 | 0.001 | 0.003 | 0.001 | 0.028 | 0.008 | 0.066 | 0.144 | 0.0196 |

## Table 3 Background input

**Background-scaling factor ($b$)**

| L2/3e | L2/3i | L4e | L4i | L5e | L5i | L6e | L6i |
|---|---|---|---|---|---|---|---|
| 1600 | 1500 | 2100 | 1900 | 2000 | 1900 | 2900 | 2100 |

(six A1 neurons) or where the spike shape was contaminated with noise because a nearby neuron often spiked close by in time (one VMC neuron). As a general word of caution, we would like to highlight that in our previous analysis of the tetrode recordings from the somatosensory cortex, separating neurons into putative inhibitory and excitatory neurons by extracellular spike shape suggested that social-touch responses of regular-spiking neurons in the somatosensory cortex changed across the estrus cycle[13]. However, in a follow-up study using juxtacellular recordings (which capture the spike shape with much higher fidelity than tetrodes), we were unable to reproduce that finding[15]. Even in brain regions where spike-shape features appear bimodal, such as S1, they are only a weak proxy for cellular identity, at least in the cortex[86].

**Statistics**. Statistical analyses were done by using built-in tests in Matlab, or by using the 'statsmodels' module (https://www.statsmodels.org) in python. All tests were two-tailed, unless explicitly stated.

**Reporting summary**. Further information on research design is available in the Nature Research Reporting Summary linked to this article.

## Data availability
The full raw data from four example neurons (the neurons shown in Fig. 3), as well as the Matlab code used to fit and analyze the statistical models and generate plots in Fig. 3 is available as supplementary material. Matlab code and a table including the response magnitudes from all fitted 'sex-touch' models, the sex and identity of the subject animal (i.e., all the data needed to reproduce Fig. 4 and 5 and reproduce the statistical modeling), is available as supplementary material. The full raw dataset is available upon request to the corresponding author.

## Code availability
All python code required to run the simulations shown in Fig. 6, and the python code to analyze the simulated data and generate the figure panels is available as supplementary material.

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

## Acknowledgements

We thank Gabriel Curio for valuable discussions. We thank Sara Helgheim Tawfiq for behavior drawings. This work was supported by The Novo Nordisk Foundation (C.L.E.), Humboldt-Universität zu Berlin within the Excellence Initiative of the states and the federal government (C.L.E.), BCCN Berlin (German Federal Ministry of Education and Research BMBF, Förderkennzeichen 01GQ1001A) (M.B.), Humboldt-Universität zu Berlin (M.B.), NeuroCure (M.B.), and the Gottfried Wilhelm Leibniz Prize of the DFG (M.B.).

## Author contributions

C.L.E., E.B., and R.P.R. provided clustered unit data and behavior data from previous studies. C.L.E. performed additional tetrode experiments. C.L.E. designed and performed analysis and statistical modeling, designed and performed spiking neural network simulations, and prepared the figures. M.B. supervised the study. C.L.E. wrote the first version of the paper. All authors contributed to writing the paper.

## Competing interests

The authors declare no competing interests.
