## [Peer Review File · Nature Communications]

Reviewers' Comments:

Reviewer #1:

Remarks to the Author:

Summary

This paper by Ebbesen et al., investigates the neural responses to social touch across many cortical areas during social touch where the recorded animals whisk and touch con-specifics of either sex across an elevated platform. The experimental procedure has been well established in the lab and this is likely a meta analyses using data collected earlier for their other studies. As shown very elegantly in their previous papers, social touch is a strong factor that modulates neuronal firing rate across areas. The behavioral paradigm is however one that is not a tightly controlled one with animals freely moving their heads when making contact and choosing different contact durations and presumably whisking strategies. A wide repertoire of behaviors accompanies this event including vocalizations as shown by the authors previously. All this might impact the data, and I think the incredible variability in the responses shown in Fig 1 bear testimony to this rich variability of responses.

Figures 1-3 are basically descriptive statistics of the data which is nice but not specifically surprising. The largest percentage of social touch and touch neurons were in fact recorded from auditory cortex (48%) which raises the question whether in fact vocalizations played a part. The fact that S1 neurons increased firing rates in response to touch and vM1 neurons decreased their firing rates is not surprising given previous literature including the authors' own M1 paper published last year. The fact that the gender of con-specifics played a part in this social touch modulation could also be inferred from Bobrov et al., 2014. The novel part of the paper is encapsulated in Figure 4 where potentiation of this response as a function of subject sex is shown in all three brain areas which is an interesting finding although no further clues for this are embedded in the data.

Major criticisms

While a rich dataset and an interesting finding, I find the presentation to be confusing with several complex transformations performed on the raw data for the main finding to be unmasked. I think these transformations should be clearly presented in a methods schematic to make the readers job easier. This is especially important for a Nature Communications with a wide readership.

This is very complex data with a huge variability across brain areas and social touch behavior episodes. The authors have given us a preliminary idea of the complexity and the interactions between the different parameters (brain area, social touch episodes, duration of contacts etc) in Figure 1. GLM modeling and log likelihood functions are a good way to deal with this variability. However, early figures should include more easy to understand PSTHs for the general readership of Nature Communications. I wonder whether the entire paper would not be more easy to grapple with if the main analyses were restricted to the cortical areas where the effects were more significant (VMC,S1,A1) and if prelimbic and cingulate cortices may not be included in the supplementary material.

PSTH windows rather large, therefore all the more difficult to look at changes....why not shorten the pre-social touch windows to exclude social touch bouts before time zero

Gaussian kernels of 100 ms width have been used to smoothen the distributions...I dont like this as this smudges out latencies and also introduces kernel shape in sparsely firing neurons. If one must use kernels for spike smoothing, an EPSP kernel is more suited

How is social touch defined? When the animal leans over and whisks against a con-specific? I imagine a lot of variability there. For example the first trials, versus later trials when the animal expected the con-specific. Were there trials where the animals whisked against the con-specific as opposed to those where it reached over without whisking and touched the other animal with extended whiskers? If so, were there differences between these situations in the same neuron (active vs passive touch). Were

there motor differences in brain areas? More quantification and explanation of behavioral data is needed to understand the study.

A1 shows the strongest proportion of social and sex touch neurons. What concerns me is whether what the authors interpret as touch neurons are touch neurons at all or social context neurons activated by any modality that indicates a social situation. Do the animals vocalize during interactions (Rao et al., *Elife*, 2014)? If so, have the authors tried looking at first pass data triggered on vocalization instead of touch? Did the authors ever try out presenting a dummy animal to their test subjects to see if similar modulations were evoked (Bobrov thesis)

Minor criticisms

The individual specific analyses (MI) while important are confusing to the main narrative and maybe should be moved to Supplementary Information

Fig 2d. Use a different arrow color instead of gray

Fig 2g. Use different colors in mosaic plot as pink and blue are the dominant sex colors throughout the paper

Page 8, Line 214: Should read 89 vs 38 neurons since 89 neurons were increased

Fig 4b. The blue and green dots are difficult to discern from one another. Please use different colors

It is not clear to me why the female modulation is the regression coefficient of touch whereas the male modulation is the sum of the coefficients for touch and sex. This should be explained better.

Reviewer #2:

Remarks to the Author:

The authors extend previous findings using tetrode recordings to assess touch- and sex-specific responses in interacting rats across cortical areas. Using a well-described social gap paradigm that the authors have previously employed in several papers, the authors here now further analyze these data sets as well as add new data sets in additional brain areas. The new data include all data from superficial layers of cortex, cingulate cortex and prelimbic cortex. Additional sex-specific response analysis has also been performed in this manuscript. Major conclusions from assessing these additional cortical areas is that touch and sex-specific touch is represented widely across cortex. The work here helps those working in this field to better appreciate sex differences in animal neurophysiological data. Other general conclusions were that these differences were best observed at the single cell level and could, in many cases, be attributed to neurons both increasing and decreasing firing rates, which could be potentially lost in population analysis. Some of the interesting aspects regarding partner sex and touch have already been reported by this laboratory in *Current Biology*, 2014.

In general, the manuscript is a neurophysiological tour-de-force. In the order of thousands of social touch episodes and even greater numbers of responses when the various brain areas are considered. My impression of the paper is that (while being very well described) statistically and from a methodological standpoint is going to be a more difficult read than similar previous Brecht lab papers on the topic. The figures will take time for most to digest and it will be a tough read for many, I don't think this is completely a fault of the authors the analysis required is quite extensive.

The papers uses many statistical models and approaches which are not necessarily in common use. An additional supplementary section which could help those not familiar with some of the statistical modeling approaches would be helpful. These enhancements might be a more clear explanation of log likelihood results in Figure 2, panels D and E. Furthermore, how log likelihood calculation is done and its role in establishing these arguments could be more clear. Furthermore, the use of pearson

residuals in Panel G is also not intuitive from what is presented the way the paper is written assumes the reader has an extensive statistical background, this could be made more accessible.

Figure 3 is also difficult to penetrate. Differences are shown in firing rate for β -touch. However, the parameter β -touch is not defined in the figure legend. β -touch is the estimated touch regression coefficient. In addition, Figure 4 which one can understand clearly from the Results section is difficult to grasp when one sees all the data plotted since there in many cases there is a huge range of a β -touch regression values. While I can see the changes within the distribution and it is appreciated how bias and potentiation are represented, it is nonetheless difficult to appreciate with so many symbols plotted and such large distributions. Perhaps there are other ways of displaying this or having a supplemental material with distribution analyses. While this is potentially something which makes the manuscript more difficult to appreciate, it's nonetheless a strong selling point for doing single cell analysis and showing these advanced statistical analyses I think the paper could be more clear and the figures made in a way to clearly show the major relationships.

Reviewer #3:

Remarks to the Author:

Ebbesen and colleagues demonstrate the rats have neurons that not only respond to social touch, but whose responses are also modulated by the gender of the animal being interacted with. The report highlights an ethologically important issue of how animals interact with each other and how these interactions drive neuronal responses. The results are novel.

Major Issues:

In the Discussion the authors point out the role that inhibitory neurons in particular can play in modulating gender specific responses, but do not analyze the waveforms of their recordings. Were fast-spiking versus regular-spiking neurons observed in the course of the recordings and did their responses vary?

Given that multiple cortical areas were probed in the same animals did specific animals show greater modulation than others? The analyses largely assume each neuron is independent from other neurons.

Given that a variety of internal physiological (e.g. hormones) vary as a function of estrous cycle. For female animals (or male animals interacting with females) did responses vary as a function of estrous cycle?

Minor Issues:

In the abstract line 31 the number of animals is indicated (n=28) but this should be reported as the number of male and the number of female animals.

In the abstract line 35 'sex-touch' responses are reported, but this term is not defined as of yet and thus its meaning is unclear.

In the Introduction the role of the Insular cortex is highlighted in sex responses, but recordings are not made from this area – why?

Do cells change their response type across trials? For example if a neuron shows increased firing in response to social touch does it always show increased firing or in some trials will it show decreased firing? Do neurons consistently respond the same way across trials or in some trials they show

modulation (increase or decrease firing) and some their firing rate remains unchanged during the social interaction? Should increasing and decreasing firing be thought as separate neurons or as part of a continuum?

What is the mechanism in which the animal is extracting the gender of the animal it is interacting with, given the illumination of the arena, is it assumed olfactory cues are mediating the responses?

the statistics appear appropriate and the methods explain the analytical methods in sufficient detail.

Berlin 18-June-2019

Thank you all for the positive comments and constructive suggestions. We are happy to now resubmit a thoroughly revised version of the manuscript. As outlined in the point-by-point response below, we think the new version of the manuscript is greatly improved, thanks to the good suggestions from the referees. The most major changes to the manuscript are along two lines. First, as suggested by all referees, we have made a major effort at **simplifying, clarifying and focusing the manuscript**. Specifically:

- (1) We have made a major effort to make the analysis methods more easily understandable for a wide audience. Specifically – as suggested by several referees – we now suggest to add a main figure, which shows the data analysis and statistical modeling ‘pipeline’ in an accessible graphical format.
- (2) We have simplified the writing of the manuscript, deleted several sections of text and removed two supplementary figures, which made the manuscript tough to read and anyway only reported ‘negative’ results. The manuscript now reads much easier and is more focused on clearly communicating our main findings, to a broad readership.
- (3) We have written an additional supplementary note, with an accessible and graphical introduction to the use of beta-coefficients as a measure of firing rate changes. As suggested by the referees, we present two complementary views of the analysis method. On the one hand, we walk the reader through a ‘classic’ PSTH-style analysis of touch responses, and show that beta-regression coefficients are just firing rate changes, but measured on a logarithmic scale. We show that this is a natural scale to use, since firing rates themselves are log-normally distributed. On the other hand, we also derive the same quantities from the modeling approach to show that a PSTH-analysis and regression are essentially two views of the same thing.

In our second major line of improvements, we are happy to also present **new scientific insights**: Based on the suggestions from the referees, we have made a major effort to understand our “main” result: that cortical networks respond to social touch with a highly structured, partner-sex and subject-sex-dependent population response. In the new version of the manuscript, we use **biologically realistic network simulations to investigate a plausible mechanism**: social-touch-dependent release of oxytocin (a prime candidate for neuromodulation of cortical inhibition during social interactions). Also, as suggested by the referees, we **investigated effects of estrus and inhibitory/excitatory neuron responses in our data**, based on spike

shape. In our simulations, **neuromodulation by oxytocin changes the network's response to thalamic input (i.e. touch) in the same way, as we found in the data** (a potentiation of responses). Inhibitory/excitatory responses predicted by the model also match our data.

We think this set of additional *in-silico* “**experiments**” and the additional **analyses suggested by the referees** add substantial scientific value to the paper, and we thank the referees for pushing us on this point. Likewise, following the suggestions to simplify writing and figures have enhanced the flow and clarity of the manuscript. We think these changes greatly improved the manuscript and we hope that the manuscript may be acceptable for publication in *Nature Communications*.

As requested, we include a supplementary data file with the raw data underlying figures and the Matlab/python code used to perform statistical modeling and analysis and perform spiking neural network simulations ('Ebbesen2019_code_and_data.7z' – if we used a regular tar or zip file, the upload system unzips all the files and creates a big mess).

One specific comment: when preparing, checking and re-running the data and code to upload and share for this re-submission, we noticed a mistake: In our shuffling test for the per-cell test for ‘sex-touch’ neurons, we mistakenly permuted both the labels (male/female partner) and the interactions – not just the labels. This a less stringent test of significance, because we destroy all correlated noise. The mistake affected only the count of ‘sex-touch’ neurons (i.e. only Fig 2g-h and 3a in the old version of the manuscript). We have fixed the mistake and when permuting only labels (the more stringent test) we report a slightly lower number of ‘sex-touch’ neurons (8.3% instead of 11%).

For readability, we repeat the referee comment in italics on the next pages, followed by our comments and changes to the manuscript. New or rewritten parts of the manuscript are highlighted.

Sincerely yours,

Christian Ebbesen

and

Michael Brecht

Reviewer #1 (Remarks to the Author):

Summary

This paper by Ebbesen et al., investigates the neural responses to social touch across many cortical areas during social touch where the recorded animals whisk and touch con-specifics of either sex across an elevated platform. The experimental procedure has been well established in the lab and this is likely a meta analyses using data collected earlier for their other studies. As shown very elegantly in their previous papers, social touch is a strong factor that modulates neuronal firing rate across areas. The behavioral paradigm is however one that is not a tightly controlled one with animals freely moving their heads when making contact and choosing different contact durations and presumably whisking strategies. A wide repertoire of behaviors accompanies this event including vocalizations as shown by the authors previously. All this might impact the data, and I think the incredible variability in the responses shown in Fig 1 bear testimony to this rich variability of responses.

Figures 1-3 are basically descriptive statistics of the data which is nice but not specifically surprising. The largest percentage of social touch and touch neurons were in fact recorded from auditory cortex (48%) which raises the question whether in fact vocalizations played a part. The fact that S1 neurons increased firing rates in response to touch and vM1 neurons decreased their firing rates is not surprising given previous literature including the authors' own M1 paper published last year.

The fact that the gender of con-specifics played a part in this social touch modulation could also be inferred from Bobrov et al., 2014. The novel part of the paper is encapsulated in Figure 4 where potentiation of this response as a function of subject sex is shown in all three brain areas which is an interesting finding although no further clues for this are embedded in the data.

COMMENT: We thank the referee for the positive assessment and constructive comments. As mentioned in the cover letter above, the high number of 'sex-touch' neurons in A1 was simply due to a bug in the fitting code (we mistakenly shuffled both the partner sex *and* the interaction times, which destroys all correlations). As evident from the updated version of the manuscript, where we have fixed this mistake, the number of sex-touch neurons in A1 is no longer larger than other areas. The referee was right to notice that the proportion of sex-touch neurons in A1 was surprising, and it was in fact a totally spurious result. Regarding the comments on the structure, we agree with the referee that the most interesting and most novel finding of our study is the former Figure 4. Namely, that across cortical areas with diverse functions (somatosensory, auditory, motor, frontal) and diverse response patterns (mostly increasing or mostly decreasing during touch), all areas shows a very similar, partner-sex-dependent and subject-sex-dependent modulation at the network level (not a shift, but a potentiation). In line with the suggestion of the referee, we have now rewritten the manuscript to concisely communicate this finding (including removing two supplementary figures and moving a former main figure to the supplementary). Moreover, in line with the comments from the referee, we have made a major effort to try to understand how this network-level modulation pattern might arise. Specifically, as prompted by comments from this referee and referee #3, we have investigated how responses map onto putatively excitatory/inhibitory neurons (assessed by spike shape), how they depend on estrus, and we have performed a set of simulations using a full-scale model of somatosensory cortex (~80k neurons, ~300 million synapses). Using our modeling, we can now report that our "main" result (the highly structured population response and potentiation) is consistent with changes in cortical inhibitory drive, and we can suggest this as a plausible mechanism.

CHANGES: Former figure 3 is now supplementary, former Figure S5 and S6 are deleted from the manuscript. We add a main figure (and three supplementary) where a biologically realistic spiking neural network model of a 1x1mm chunk of somatosensory cortex shows the same modulation pattern as we observed experimentally, when we simulate the neuromodulatory effect of oxytocin release during social touch.

Major criticisms

While a rich dataset and an interesting finding, I find the presentation to be confusing with several complex transformations performed on the raw data for the main finding to be unmasked. I think these

transformations should be clearly presented in a methods schematic to make the readers job easier. This is especially important for a Nature Communications with a wide readership.

COMMENT: We agree very much with the comment of the referee.

CHANGE: As suggested, we have added a methods schematic (new Figure 2), which presents the analysis 'pipeline' (from raw data to fitted models) in a graphical format, clearly and accessible for a wide readership.

This is very complex data with a huge variability across brain areas and social touch behavior episodes. The authors have given us a preliminary idea of the complexity and the interactions between the different parameters (brain area, social touch episodes, duration of contacts etc) in Figure 1. GLM modeling and log likelihood functions are a good way to deal with this variability. However, early figures should include more easy to understand PSTHs for the general readership of Nature Communications.

COMMENT: We thank the referee for the positive assessment of our analysis method. We agree with the referee that while the analysis is well-suited to analyze complex data, the methods are more "math-heavy" than other types of analyses, commonly used in the field. We agree with the suggestion from the referee, that comparing/contrasting our analysis method with a 'classic' PSTH-style analysis is a very good idea. That type of analysis is familiar in the field, and comparing our method directly to such an analysis makes the paper accessible to a much wider readership.

CHANGE: We have written a supplementary note, where we walk the reader through a 'classic' PSTH-style analysis of touch responses, compare that an analysis to our approach and shows that our modeling and beta-regression coefficients are really just firing rate changes, but measured on a logarithmic scale. We explain that his logarithmic scale is a natural choice, since firing rates are log-normally distributed (Buzsáki and Mizuseki, 2014; Silver, 2010). This supplementary note also makes heavy use of schematics/drawings to make the material easily accessible and intuitive to a readership, which is more familiar with analyzing touch responses using PSTHs. In line with this, we have also added sentences throughout the text, which clearly explain with words how to think about beta-coefficients, for example, see the updated text in L. 155-164.

I wonder whether the entire paper would not be more easy to grapple with if the main analyses were restricted to the cortical areas where the effects were more significant (VMC, S1, A1) and if prelimbic and cingulate cortices may not be included in the supplementary material.

COMMENT: Like the referee, we also wondered if we should simply leave out the Cg1 data and the PrL data, to simplify the paper/figures. However, even though the network-level effects are not significant in these areas (we think, simply due to the lower N), the maximum-likelihood estimates are all in the same direction as for VMC, S1 and A1. This did not have to be the case. From a holistic, Bayesian perspective, the fact that even in the areas with a lower N, the pattern is similar, this adds even more confidence to our reported findings. For this reason, we think it would be a shame to totally delete the data from the paper. CHANGE: We have decided on a compromise: The old figure 3 is now supplementary, so we do not walk the reader through that data from ACC and PrL, we simply show it in the supplementary data. We think this was a very good suggestion. Now data from ACC and PrL are only shown in Figure 1 (Main

New Main Figure 2: Graphical description of the analysis (see manuscript)

New Suppl. Note 1: Accessible introduction to beta-coefficients, with graphical comparison to a 'classic' PSTH-style analysis (see manuscript).

message: Touch responses are diverse, and found in all the areas that we recorded from, even ACC and PrL) and Figure 3 (Main message: Touch and Sex-Touch responses are widely distributed, found in all areas that we recorded from, even ACC and PrL).

Gaussian kernels of 100 ms width have been used to smoothen the distributions...I dont like this as this smudges out latencies and also introduces kernel shape in sparsely firing neurons. If one must use kernels for spike smoothing, an EPSP kernel is more suited

COMMENT: We agree with the referee that a PSP-shaped kernel is actually a much better choice for smoothing the PSTHs.

CHANGE: We have changed the smoothing kernel in all the figures, from a Gaussian with $\sigma = 100$ ms to a more narrow, and PSP-shaped alpha function with $\tau = 75$ ms.

Ref. Figure 1: New Kernel

PSTH windows rather large, therefore all the more difficult to look at changes....why not shorten the pre-social touch windows to exclude social touch bouts before time zero

COMMENT: As the referees also points out elsewhere, there is a large variability from trial to trial. The point of Figure 1 is simply to transparently show this variability, in a raw and 'naïve' way. We suggest keeping all trials there, for easier comparison with Figure 3, which also has all trials, but now sorted by sex and with all touches in the 'baseline' labeled. All 'actual' analysis is anyway done using the GLM approach. Thus, Figure 1 only shows example neurons and serves a didactic point, namely to show the advantages of using the GLM to handle the complexity of the data, and optimally estimate (within the MLE model, anyway) modulation by touch, despite the variability in onset times, durations, etc. For the former Figure S5 (where we looked for onset of the responses), the comment of the referee is a valuable suggestion. In that analysis, we previously already excluded any trials where there was a touch bout before zero, and used as short touch windows as seemed statistically stable. Obviously, with a PSP-shaped kernel, this window length could in principle have changed and made shorter windows feasible. As pr. the comment above, we repeated the analysis again, this time with both shorter and longer windows and with the PSP-shaped smoothing. We show this figure as Referee Fig 2 below, with longer windows to show the variability. As is evident from the example neurons, removing the touch bouts before zero does not get rid of the variability, and there is simply too much variability at short time scales to make really short analysis windows feasible. To reliably estimate (despite the variability) when the PSTHs are modulated, we have to look at longer time scales. Actually, in the end, in line with the suggestion from several referees to simplify the paper, and focus the paper on the major novel findings, we simply decided to remove this figure and discussion from the manuscript.

Referee Fig. 2: Estimating onsets, with PSP kernel and a longer window (which – to simplify the paper, as suggested – we have removed from the manuscript)

CHANGE: We re-analyzed the onset times, with the suggested PSP kernel and both shorter and longer windows (shown below). However, in line with the suggestion from several referees to simplify the paper, and focus the paper on the major novel findings, we actually decided in the end to rather simply remove this figure and discussion from the manuscript.

How is social touch defined? When the animal leans over and whisks against a con-specific? We defined social touch as from the first whisker-to-whisker contact to the last whisker-to-whisker contact, as described in the methods section.

I imagine a lot of variability there. For example the first trials, versus later trials when the animal expected the con-specific.

Like the referee, we would be very interested in understanding how the male/female differences arise. Are they learned slowly over time, innate, modulated by puberty? The data in this paper cannot speak to these questions, since – as mentioned in the methods section – all rats, both subjects and partners, were already well-habituated to the experimental setup and to each other (the two platforms, reaching to touch each other in darkness) when we started the recording sessions. There is no element of learning across our recordings. There also isn't an element of surprise or expectation, at shorter time scales. For a social interaction to occur, both rats have to reach across the gap. Thus, the interactions are very much jointly initiated. It is not the case, for example, that one rat initiates the interaction, and the other follows or expects/doesn't expect it. In previous behavioral studies (Wolfe et al., 2011), we actually looked for exactly this, since we were wondering if, for example, dominant animals would 'initiate' an interaction. Presumably, the animals jointly align and initiate and both contribute by emitting and sensing olfactory and auditory cues.

Were there trials where the animals whisked against the con-specific as opposed to those where it reached over without whisking and touched the other animal with extended whiskers? If so, were there differences between these situations in the same neuron (active vs passive touch). Were there motor differences in brain areas?

In the previous study, focusing on S1 (Bobrov et al., 2014), we asked ourselves the same question. The whisker movements are very similar on average, but they do vary from interaction to interaction. In that previous study, we used high-speed video to track single whiskers and found that subject and stimulus rat whisking was not correlated or only very weakly correlated with neuronal responses. In a follow-up study, investigating social facial touch in head-fixed rats (Lenschow and Brecht, 2015), we also tracked single whiskers and also did not find that whisker movements explained social touch responses in S1: membrane potential modulation during both social touch and free whisking did not correlate with whisking parameters.

Motor differences: Because the animals are freely moving, orienting and aligning during social facial touch episodes, it was not possible for us to automatically track motor parameters, such as whisking amplitudes, in the free interactions. In a previous study, focusing on VMC (Ebbesen et al., 2017), we investigated if social touch responses in VMC could be explained simply by motor differences, by recording from head-fixed, socially interacting animals, where whisker tracking was possible. In these experiments we found that differences in whisker movements did not explain the social touch responses. We used a GLM approach to regress social touch and movement simultaneously and found that even when we controlled for whisking kinematics, there was still an additional and large effect of social touch. It is an aspect of our study, that while we have a very large dataset of neural data, these raw data are stemming from multiple previous studies, with different foci. We sadly do not have high-speed single-whisker tracking, whisker movement data and ultrasonic call data available for all cells, across all interactions, across all brain areas.

A1 shows the strongest proportion of social and sex touch neurons. What concerns me is whether what the authors interpret as touch neurons are touch neurons at all or social context neurons activated by any modality that indicates a social situation. Do the animals vocalize during interactions (Rao et al., *Elife*, 2014)? If so, have the authors tried looking at first pass data triggered on vocalization instead of touch? Did the authors ever try out presenting a dummy animal to their test subjects to see if similar modulations were evoked (Bobrov thesis)

COMMENT: We fully agree with this important point. This was clearly imprecise wording on our part, and which we are grateful to the referee for pointing out. We absolutely do *not* wish to suggest that the neurons which display 'touch' or 'sex-touch' responses in our paradigm are activated only by whisker touch and not by any other sensory modality or in any other situation. For example, as pointed out by the referee, neurons in A1 are modulated by both touch and calls (Rao et al., 2014). Like the referee, we are not fans of categorical classification of rat neurons into cell 'types', based on $p < 0.05$, an arbitrary threshold imposed by human biologists. We merely present this data, classified in types, since this is an intuitive way to quantify how 'common' a type of response is across a given network. This is also why, in our main effect, we do not restrict the analysis, but analyze all neurons in each area jointly and characterize the whole network response (to avoid statistical pitfalls like Berkson's paradox). Like the referee suggests, we also prefer to remain totally agnostic about what drives these sensory responses during social facial touch. When we refer to neurons as 'touch neurons' for example, we simply mean that these are neurons which display firing rate changes during social touch. We do not wish to suggest that these neurons are driven purely by touch, and not also by calls, olfactory cues, temperature, or other internal processes. Social touch is clearly multisensory, and we also think that the "social" responses are probably multisensory.

Wrt. to presenting a stuffed animal, like in the Bobrov thesis or like in Lenschow & Brecht 2015: Yes, we

Referee Figure 3: S1 responses to a stuffed rat. (a) A taxidermied rat, presented during experiments. (b) Examples of real social facial touch, and whisker touch with the stuffed rat (tetode implant digitally removed). (c) Number of touch episodes with stuffed conspecifics and real rats, across recording sessions. (d) Joint distribution of touch modulation with real and stuffed conspecifics: where the rates are estimated in 500-ms windows around touch onset. (e) Density of touch responses, plotted as histograms. Distribution of responses to stuffed animals is wider, probably due to the much lower N. (f) Touch responses to conspecifics are bigger than responses when touching a stuffed rat ($p = 0.0016$, paired t -test) (g) The distribution of paired differences is significantly below zero (and roughly normal, cf. the t -test).

did pilot such experiments both in S1, A1 and VMC (Referee Fig. 3a-b), to investigate sensory responses to a “rat-shaped” object. In contrast to in the head-fixed prep (Lenchow and Brecht, 2015), where we can present the stuffed animal repeatedly, we could only collect very few “interactions” with the stuffed animals in this freely moving paradigm. Basically, the rats do not seem to care much about the stuffed animals, and just (from observing and anthropomorphizing the animals a little bit), they seem to treat them like any other object. They are not afraid of them, for example, they just do not seem to recognize them as (taxidermied) conspecifics. Thus, if we place a stuffed animal on one platform and let the rats “interact”, they will generally go and touch the stuffed animal only a few times, or when habituated, often not at all (it is in darkness, after all, and sometimes the rats will prefer to groom/sleep, when they are alone, instead of whisking onto a well-known object, to which they have been habituated).

Based on the comments from the referee, we looked into the data, and compared responses estimated by comparing the firing rate in a 500-ms window before and after the beginning of the interaction. Since we have much fewer ‘interactions’ with a stuffed animal than with conspecifics (Ref. Fig 3c), there was a large scatter in the estimate of the modulation with the stuffed animal (Ref Fig. 3d-e, distribution of stuffed-rat indices is very wide). Nonetheless, in S1, we found that responses to touching live conspecifics were significantly larger than responses to stuffed animals ($p = 0.0016$, paired t -test, distribution of $\beta_{stuffed} - \beta_{touch}$ below zero, Ref. Fig. 3f-g). In VMC (where we have good data coverage) and A1 (where we have little data) we did not find a significant difference (Ref. Fig 4).

Referee Figure 4: Responses to a stuffed rat in A1 and VMC are not different from responses to live conspecifics ($p \gg .05$ for both). (a-d) Same as panel a,d,e,g in previous figure, but for data from VMC. (e-h) Same as panel a,d,e,g in previous figure, but for data from A1.

We quite like these data, since – as the referee suggested – they are consistent with the idea that responses during social touch are multisensory. However, while rats whisk very similarly onto male and female conspecifics, they whisk quite differently onto conspecifics and objects. Social touch is quite delicate, involves complex movements and under fine motor control, whereas object/stuffed rat touch consists of larger, more stereotyped whisks. For that reason, when we compare whisking onto stuffed animals to real social whisking, we are comparing very different whisking patterns, thus we do not want to make any strong claims on this point. However, in general, we agree with the referee that this is an interesting question.

CHANGE: We have rewritten the text, so that it is clear from the reader that we are not suggesting that ‘touch’ neurons are only activated by touch, and nothing else. From our experiment, we cannot not know what “really” drives these neurons. We simply mean that, in a purely descriptive sense, these neurons are modulated during episodes of social facial touch. We point out again in the discussion, that the responses during social touch is presumably highly multisensory. We note that “*Our definition of social ‘touch’ neurons is purely descriptive and we do not know what low-level features of the social touch episode (touch, ultrasonic calls, olfactory cues, temperature) or internal ‘top-down’ processes modulate the firing rate of these neurons.*” (line 455-457)

Minor criticisms

The individual specific analyses (MI) while important are confusing to the main narrative and maybe should be moved to Supplementary Information

COMMENT & CHANGE: We agree. We have moved this to the supplementary.

Fig 2d. Use a different arrow color instead of gray

COMMENT & CHANGE: We have changed this color.

Fig 2g. Use different colors in mosaic plot as pink and blue are the dominant sex colors throughout the paper

COMMENT & CHANGE: We have changed the color.

Page 8, Line 214: Should read 89 vs 38 neurons since 89 neurons were increased

COMMENT & CHANGE: We have fixed this, and this part is only supplementary now.

Fig 4b. The blue and green dots are difficult to discern from one another. Please use different colors. It is not clear to me why the female modulation is the regression coefficient of touch whereas the male modulation is the sum of the coefficients for touch and sex. This should be explained better.

COMMENT & CHANGE: The choice to have male as 1 and female as 0 on the indicator variable (so that in the sex-touch models, the default 'touch' corresponds to female) is arbitrary, it could also have been the other way around, and does not affect the fit/results. We thank the referee for pointing out that there is potential for confusion here. We have improved the description, both in the main text and in the methods sections. Especially, the new Figure 2 helps in this regard. Moreover, we have added an accessible explanation and a few worked-through examples to the new supplementary note, aimed at readers that are not so familiar with this type of analysis. We think that this makes it clearer.

Reviewer #2 (Remarks to the Author):

The authors extend previous findings using tetrode recordings to assess touch- and sex-specific responses in interacting rats across cortical areas. Using a well-described social gap paradigm that the authors have previously employed in several papers, the authors here now further analyze these data sets as well as add new data sets in additional brain areas. The new data include all data from superficial layers of cortex, cingulate cortex and prelimbic cortex. Additional sex-specific response analysis has also been performed in this manuscript. Major conclusions from assessing these additional cortical areas is that touch and sex-specific touch is represented widely across cortex. The work here helps those working in this field to better appreciate sex differences in animal neurophysiological data. Other general conclusions were that these differences were best observed at the single cell level and could, in many cases, be attributed to neurons both increasing and decreasing firing rates, which could be potentially lost in population analysis. Some of the interesting aspects regarding partner sex and touch have already been reported by this laboratory in Current Biology, 2014.

COMMENT: We thank the referee for the very positive assessment of our study.

In general, the manuscript is a neurophysiological tour-de-force. In the order of thousands of social touch episodes and even greater numbers of responses when the various brain areas are considered. My impression of the paper is that (while being very well described) statistically and from a methodological standpoint is going to be a more difficult read than similar previous Brecht lab papers on the topic. The figures will take time for most to digest and it will be a tough read for many, I don't think this is completely a fault of the authors the analysis required in quite extensive.

COMMENT: We agree that the previous version of the manuscript was hard to digest.

CHANGE: In the new version of the manuscript, we have made a major effort to make the paper easier to read. We add a graphical figure showing the analysis pipeline (Fig. 2), and we have added an accessible supplementary note, where we graphically introduce the methods. Please also see our comments to similar points raised by referee #1.

The papers uses many statistical models and approaches which are not necessarily in common use. An additional supplementary section which could help those not familiar with some of the statistical modeling approaches would be helpful. These enhancements might be a more clear explanation of log likelihood results in Figure 2, panels D and E. Furthermore, how log likelihood calculation is done and its role in establishing these arguments could be more clear. Furthermore, the use of pearson residuals in Panel G is also not intuitive from what is presented the way the paper is written assumes the reader has an extensive statistical background, this could be made more accessible.

COMMENT: We agree with the comments from the referee, which are very similar to the points raised by referee #1, and to the comment above.

CHANGE: We have added a new Figure 2 and a new Supplementary note 1. As suggested by the referee, we have added sentences explaining the Pearson residuals, to a reader which might be unfamiliar with the measure. If a χ^2 -test rejects the hypothesis that a multi-dimensional contingency table is randomly distributed, that does not tell us which proportions are deviating from random. The standardized Pearson residuals pr. square in the mosaic plot is just a simple way to illustrate which proportions deviate from a random model (the residuals are standardized, such that they follow a standardized normal distribution, and are easy to interpret, for example a standardized residual $> +/- 1.96$ corresponds to $p < 0.05$).

Figure 3 is also difficult to penetrate. Differences are shown in firing rate for β -touch. However, the parameter β -touch is not defined in the figure legend. β -touch is the estimated touch regression coefficient. In addition, Figure 4 which one can understand clearly from the Results section is difficult to grasp when one sees all the data plotted since there in many cases there is a huge range of a β -touch regression values. While I can see the changes within the distribution and it is appreciated how bias and potentiation are represented, it is nonetheless difficult to appreciate with so many symbols plotted and such large distributions. Perhaps there are other ways of displaying this or having a supplemental material with distribution analyses. While this is potentially something which makes the manuscript more difficult to appreciate, it's nonetheless a strong selling point for doing single cell analysis and showing these advanced statistical analyses I think the paper could be more clear and the figures made in a way to clearly show the major relationships.

COMMENT: We agree with the comments of the referee. These points were also raised by referee #1. Please also see our comments to referee #1 above.

CHANGE: In the spirit of simplifying the paper, we have moved the former figure 3 to the supplementary material, and provide an easier-to-digest summary of the results of the figure in the main text.

Reviewer #3 (Remarks to the Author):

Ebbesen and colleagues demonstrate the rats have neurons that not only respond to social touch, but whose responses are also modulated by the gender of the animal being interacted with. The report highlights an ethologically important issue of how animals interact with each other and how these interactions drive neuronal responses. The results are novel.

COMMENT: We thank the referee for appreciating our contribution.

Major Issues:

In the Discussion the authors point out the role that inhibitory neurons in particular can play in modulating gender specific responses, but do not analyze the waveforms of their recordings. Were fast-spiking versus regular-spiking neurons observed in the course of the recordings and did their responses vary?

COMMENT: We are very grateful to the referee for this comment. In the previous version of the discussion, we suggested that our main finding (the network-level response potentiation, which signals both partner sex and subject sex) might suggest that social touch is associated with a modulation of inhibitory drive as we suggested, for example by oxytocin. Oxytocin is known to be released during mammalian social touch, there are direct fibers from the PVN to cortex, the receptor is expressed by cortical interneurons, and oxytocin has been shown to modulate inhibition in slices. The referee is right to point out that we could push this angle more, and that there might be more interesting things to discover. Based on the comment from the referee, we did several things:

- (1) First, we clarified our previous discussion by sketching out the effect we had in mind and were hinting at in the discussion: if feed-forward inhibition is modulated, for example by oxytocin, we can easily imagine circuit motifs, which would be consistent with our observed network response pattern. For example, if suppressed VMC excitatory neurons are suppressed by inhibitory interneurons during touch (as we suggested in Ebbesen et al. 2017, based on patch experiments), increasing the activity of the interneuron would potentiate the response of both neurons, consistent with what we saw in our VMC data (Referee Figure 5a). Similarly, we can come up with circuit motifs, where modulation of feed-forward inhibition can increase responses in excitatory neurons (by increasing the

activity of I_1 in Referee Figure 5b). Obviously, we can come up with a circuit motif consistent with almost any modulation pattern, so this is only

a very weak test of our ideas. Thus, we turned to a more challenging test:

- (2) Reasoning about how modulation of inhibition might flow through a tiny motif is one thing, extrapolating that to a whole cortical population is something very different. In a realistic circuit, there will be a variety of motifs: feed-back, feed-forward, recurrent, lateral, etc. It is impossible to reason about how modulation of inhibition will change response patterns in such a complicated network – here, we have to turn to simulations. We are in the fortunate situation that previous work has developed a large-scale spiking neural network model of primary sensory cortex (Potjans and Diesmann, 2014). This model allows us to simulate the activity of $\sim 1 \times 1$ mm of

Ref. Figure 5: Circuit motifs, where a change in feed-forward inhibition would modulate population activity similarly to what we observed in the data.

somatosensory cortex (~80,000 modeled neurons, with ~ 300 million synapses) with realistic synaptic connectivity and microcircuit patterns drawn from functional and anatomical studies of S1/V1 (Potjans and Diesmann, 2014), the model has been validated several times and reproduces cell-type-specific and layer-specific distributions of firing rates, correlations, spike delays, etc. (van Albada et al., 2018; Cain et al., 2016; Potjans and Diesmann, 2014; Shimoura et al., 2018)

The effect of oxytocin on cortical circuits has been investigated in several studies, and across both cortical areas and the hippocampus, two effects has been observed, one on interneurons and on excitatory neurons: (1) Depolarization and consequently increased firing rate of inhibitory interneurons, with no change in IPSC amplitude (Marlin and Froemke, 2016; Marlin et al., 2015; Mitre et al., 2016; Nakajima et al., 2014; Owen et al., 2013) (2) An increase in input resistance on excitatory neurons. (Rogers-Carter et al., 2018; Tirko et al., 2018).

We decided to use this validated model of primary sensory cortex to simulate the effect of oxytocin action on cortical circuits during social touch. We made several discoveries:

- Increasing inhibitory drive by depolarizing interneurons lead to a potentiation, just as we observed in our data (Fig 5.c, first column: slope different from unity, no shift).
 - Increasing the input resistance of excitatory neurons simply lead to a bias in responses (Fig 5.c, first row: upwards shift from unity line).
 - When both effects are applied simultaneously, the effects interestingly “cancel out” (Fig 5.c, diagonal). We reason that (1) Inhibitory modulation – for example by oxytocin – is a plausible candidate mechanism to explain the population response modulation, we describe across multiple areas. (2) Increased input resistance of excitatory neurons, such as observed after sustained bath application of oxytocin, might be a homeostatic response to “re-normalize” network activity in answer to oxytocin-mediated changes in inhibition.
- (3) We also looked at the spike shapes of our recorded neurons, to see if putatively inhibitory/excitatory neurons respond in a manner consistent with the model. In S1 (New Suppl. Fig. 7, shown above), as well as in A1 and VMC, we found that responses of putatively inh./ex. neurons generally respond very similarly, and in the same directions. There was no indication, that interneurons and excitatory neurons respond in opposite directions, for example. The large overlap in response magnitudes, and the similar direction firing rate modulation of inh./ex. responses are in agreement with the prediction of the model (New Suppl. Fig. 7, panel a, shown above).

CHANGE: We add a new main Figure 5 and S7

New Main Figure 5 (left) and part of Suppl. Figure S7 (right). Based on suggestion from the referee, we more deeply investigated the potential role of inhibition in modulating the network pattern. Main messages: (Figure 5): Simulating neuromodulation by oxytocin release during social touch (depolarizing the interneurons) in a full-scale network model of S1 changes the network's response to thalamic input (i.e. touch) in the same way, as we find the data (note first column of panel c). (Figure SX): If we split neurons into putative inhibitory and excitatory neurons, both populations respond in the same direction and overlap (panels b-d), which is also consistent with what the modeling predicts (panel a). See manuscript for details and figure legend.

Given that a variety of internal physiological (e.g. hormones) vary as a function of estrous cycle. For female animals (or male animals interacting with females) did responses vary as a function of estrous cycle?

COMMENT: We only have the estrus state of the subject animals available in the data from S1. Reassuringly, though, we did not see a dependence of the responses (which are the focus of our in this paper) on estrus state, in either putatively inhibitory or excitatory neurons. We also did not find any differences in social touch responses across estrus states in head-fixed animals in a recent study (Clemens et al., 2019). The original paper on S1 (Bobrov et al., 2014) reported that ongoing firing rates were lower in estrus than in non-estrus (consistent with our more recent report showing increased ongoing firing rates of putatively inhibitory neurons during estrus, Clemens et al. 2019). Bobrov et al. 2014 did investigate if the responses change with estrus, using a PSTH-based approach. Using our more sophisticated analysis on the same data, we first found that baseline firing rates (measured as β_0 , so baseline rate $r_{base} = \exp(\beta_0)$, Ref. Fig 7a) were lower in estrus than non-estrus. We both did a "raw" t-test where we pooled all non-estrus states (like in the original study) and tested against the estrus state cells ($p = 0.0017$), and a more sophisticated model, where we treat all non-estrus days (pro-estrus, met-estrus and di-estrus) as independent categorical variables, control for unequal number of neurons recorded on the same experimental session, fit a GLME regression and do the full ANOVA ($p = 0.018$).

Both were significant. In contrast, when we did the same analysis on the responses (β_{touch} , Ref. Fig 7b), we did not find any difference (both t-test and GLME model: $p \gg 0.05$, same finding as Clemens et al. 2019). We tried a variety of models looking if responses depended on estrus state ('cycle', 4 levels) or putative inh./ex. neuron type ('spike_shape', 2 levels, shown in Ref. Fig. 7c), or both, but none of these models had significant effects or did better than a constant model (models compare by BIC, shown in Ref. Fig. 7d). While there surely is some variability in the female responses – in line with what one generally finds (Becker et al., 2016; Prendergast et al., 2014) – we do not see an indication that hormonal cycling makes the female data obviously more variable than the data from males, or that our reported findings are artefacts of a hidden estrus variable, strongly impacting the data. If responses are estrus dependent, then that effect is subtle and it is not possible for us to detect it in the data.

New Fig S6: In our data from S1, ongoing firing rates vary with estrus state, but responses do not change. It seems like the network responds similarly across estrus states.

Given that multiple cortical areas were probed in the same animals did specific animals show greater modulation than others? The analyses largely assume each neuron is independent from other neurons.
COMMENT: We did not see any significant differences or obvious differences visible to the eye between neurons recorded in different subject animals. Obviously, when we slice the data by animals, the number of cells becomes low and that could in principle be a reason why we see no differences between animals. Thus, we already decided to include a grouped error term (a (1|rat) term) in the GLME modeling in Figure 4 (where we are exactly looking at differences between male and female subject animals), just to be safe and to make sure that observed sex differences are not just artifacts due to prominent, random between-animal variation in response magnitude. While we stated this in the methods section, we apologize that this was not more clear from reading the manuscript. We will add a sentence to the main text to point this out.

CHANGE: We have made it more prominent that the statistical modeling in Fig 4 controls for unequal number of neurons recorded in the subject animals.

Minor Issues:

In the abstract line 31 the number of animals is indicted (n=28) but this should be reported as the number of male and the number of female animals.

COMMENT & CHANGE: We have added these numbers. Actually, we are embarrassed to report that the 28 rats were originally a typo. It is in fact 29 rats, 14 males and 15 females.

In the abstract line 35 'sex-touch' responses are reported, but this term is not defined as of yet and thus its meaning is unclear.

COMMENT & CHANGE: We agree. We have changed the sentence to: "Across cortex, 25.7% of single neurons were modulated during social touch and 11.9% of single neurons displayed 'sex-touch' responses (responded differently to touch, depending on the sex of the interaction partner)." (and we have generally essentially rewritten the abstract)

In the Introduction the role of the Insular cortex is highlighted in sex responses, but recordings are not made from this area – why?

COMMENT: Our interest in the insular cortex is a bit post-hoc. Based on reading in particular the human neuroscience literature, we have come to think that (in addition to the cortical areas investigated here)

insular cortex would also be an interesting structure to investigate during social touch. Despite the fact that we sadly do not have any data from this region, we still think it is correct to mention this area in the introduction, so as to give a broad and balanced overview of the field across both the rodent and human literature.

Do cells change their response type across trials? For example if a neuron shows increased firing in response to social touch does it always show increased firing or in some trials will it show decreased firing? Do neurons consistently respond the same way across trials or in some trials they show modulation (increase or decrease firing) and some their firing rate remains unchanged during the social interaction? Should increasing and decreasing firing be thought as separate neurons or as part of a continuum?

COMMENT: We think that the neurons should probably be thought of as a continuum, for several reasons: ● While they do have long tails, none of the distributions of touch modulation (Old Fig 3, now moved as a supplementary figure) appear obviously bimodally distributed, and do not indicate separate populations of increasing and decreasing neurons. ● In the previous version of the manuscript, we also looked if there were separate ‘transient’ and ‘sustained’ responses, but found a continuum from quite transient to tonically active response types. (in order to simplify the manuscript, we have removed this from the current version) ● As outlined in Figure 4, it is not the case that single neurons have significant responses to male/female partners with no population structure (as have been described in the PPC, (Raposo et al., 2014)). Rather, the single ‘significant’ neurons appear to be part of a highly structured, partner-sex and subject-sex-dependent population response (Figure 4). An intuitive way to get at the question of the referee – if neurons always show increases or sometimes ‘change direction’ – is to look at the distribution of neurons in Figure 4c and 4d, for example. Some neurons are in the first and third quadrant (responding in the same direction with both males and female) while some are in the second and fourth quadrants (responding in different directions, increasing with males and decreasing with females, for example). However, when we view all neurons together, it looks more like a population response, which is ‘turned away’ from the unity line, rather than distinct separate clusters. If the neurons end up exactly on this or that side of the lines separating the quadrants is probably determined by how they relate to the population response structure, and this relation is probably what ‘actually’ matters in terms of coding (Saxena and Cunningham, 2019).

What is the mechanism in which the animal is extracting the gender of the animal it is interacting with, given the illumination of the arena, is it assumed olfactory cues are mediating the responses?

COMMENT: Yes, we also think that rats mainly rely on olfactory cues in order to determine the sex of their interaction partner. Still, other sensory modalities might also contribute. For example, there might be subtle sex-differences in whisking patterns (Wolfe et al., 2011) or vocalization patterns. We know that rats vocalize around and during social facial interactions (Rao et al., 2014) and that rat pups (Bowers et al., 2013) and mice (Warren et al., 2018) display subtle sex differences in vocalizations.

the statistics appear appropriate and the methods explain the analytical methods in sufficient detail.

COMMENT: We thank the referee for the positive assessment.

References

- van Albada, S.J., Rowley, A.G., Senk, J., Hopkins, M., Schmidt, M., Stokes, A.B., Lester, D.R., Diesmann, M., and Furber, S.B. (2018). Performance comparison of the digital neuromorphic hardware SpiNNaker and the neural network simulation software NEST for a full-scale cortical microcircuit model. *Front. Neurosci.* *12*, 1–20.
- Becker, J.B., Prendergast, B.J., and Liang, J.W. (2016). Female rats are not more variable than male rats: A meta-analysis of neuroscience studies. *Biol. Sex Differ.* *7*, 1–7.
- Bobrov, E., Wolfe, J., Rao, R.P., and Brecht, M. (2014). The representation of social facial touch in rat barrel cortex. *Curr. Biol.* *24*, 109–115.
- Bowers, J.M., Perez-Pouchoulen, M., Edwards, N.S., and McCarthy, M.M. (2013). Foxp2 Mediates Sex Differences in Ultrasonic Vocalization by Rat Pups and Directs Order of Maternal Retrieval. *J. Neurosci.* *33*, 3276–3283.
- Buzsáki, G., and Mizuseki, K. (2014). The log-dynamic brain: How skewed distributions affect network operations. *Nat. Rev. Neurosci.* *15*, 264–278.
- Cain, N., Iyer, R., Koch, C., and Mihalas, S. (2016). The Computational Properties of a Simplified Cortical Column Model. *PLoS Comput. Biol.* *12*, e1005045.
- Clemens, A.M., Lenschow, C., Beed, P., Li, L., Sammons, R., Naumann, R.K., Wang, H., Schmitz, D., and Brecht, M. (2019). Estrus-Cycle Regulation of Cortical Inhibition. *Curr. Biol.* *29*, 605-615.e6.
- Ebbesen, C.L., Doron, G., Lenschow, C., and Brecht, M. (2017). Vibrissa motor cortex activity suppresses contralateral whisking behavior. *Nat. Neurosci.* *20*, 82–89.
- Lenschow, C., and Brecht, M. (2015). Barrel Cortex Membrane Potential Dynamics in Social Touch. *Neuron* *85*, 718–725.
- Marlin, B.J., and Froemke, R.C. (2016). Oxytocin modulation of neural circuits for social behavior. *Dev. Neurobiol.*
- Marlin, B.J., Mitre, M., D'amour, J.A., Chao, M. V., and Froemke, R.C. (2015). Oxytocin enables maternal behaviour by balancing cortical inhibition. *Nature* *520*, 499–504.
- Mitre, M., Marlin, B.J., Schiavo, J.K., Morina, E., Norden, S.E., Hackett, T.A., Aoki, C.J., Chao, M. V, and Froemke, R.C. (2016). A Distributed Network for Social Cognition Enriched for Oxytocin Receptors. *J. Neurosci.* *36*, 2517–2535.
- Nakajima, M., Görlich, A., Heintz, N., Anacker, A.M., Beery, A.K., Ascoli, G.A., Alonso-Nanclares, L., Anderson, S.A., Barrionuevo, G., Benavides-Piccione, R., et al. (2014). Oxytocin modulates female sociosexual behavior through a specific class of prefrontal cortical interneurons. *Cell* *159*, 295–305.
- Owen, S.F., Tuncdemir, S.N., Bader, P.L., Tirko, N.N., Fishell, G., and Tsien, R.W. (2013). Oxytocin enhances hippocampal spike transmission by modulating fast-spiking interneurons. *Nature* *500*, 458–462.
- Potjans, T.C., and Diesmann, M. (2014). The Cell-Type Specific Cortical Microcircuit: Relating Structure and Activity in a Full-Scale Spiking Network Model. *Cereb. Cortex* *24*, 785–806.
- Prendergast, B.J., Onishi, K.G., and Zucker, I. (2014). Female mice liberated for inclusion in neuroscience and biomedical research. *Neurosci. Biobehav. Rev.* *40*, 1–5.
- Rao, R.P., Mielke, F., Bobrov, E., and Brecht, M. (2014). Vocalization–whisking coordination and multisensory integration of social signals in rat auditory cortex. *Elife* *3*, 1–20.

- Raposo, D., Kaufman, M.T., and Churchland, A.K. (2014). A category-free neural population supports evolving demands during decision-making. *Nat. Neurosci.* *17*, 1784–1792.
- Rogers-Carter, M.M., Varela, J.A., Gribbons, K.B., Pierce, A.F., McGoey, M.T., Ritchey, M., and Christianson, J.P. (2018). Insular cortex mediates approach and avoidance responses to social affective stimuli. *Nat. Neurosci.* *21*, 1–11.
- Saxena, S., and Cunningham, J.P. (2019). Towards the neural population doctrine. *Curr. Opin. Neurobiol.* *55*, 103–111.
- Shimoura, R.O., Kamiji, N.L., Pena, R.F.O., Cordeiro, V.L., Ceballos, C.C., Romaro, C., and Roque, A.C. (2018). [Re] The cell-type specific cortical microcircuit: relating structure and activity in a full-scale spiking network model. *ReScience* *4*, 1–12.
- Silver, R.A. (2010). Neuronal arithmetic. *Nat. Rev. Neurosci.* *11*, 474–489.
- Tirko, N.N., Eyring, K.W., Carcea, I., Mitre, M., Chao, M. V., Froemke, R.C., and Tsien, R.W. (2018). Oxytocin Transforms Firing Mode of CA2 Hippocampal Neurons. *Neuron* *100*, 593-608.e3.
- Warren, M.R., Spurrier, M.S., Roth, E.D., and Neunuebel, J.P. (2018). Sex differences in vocal communication of freely interacting adult mice depend upon behavioral context. *PLoS One* *13*, 1–22.
- Wolfe, J., Mende, C., and Brecht, M. (2011). Social facial touch in rats. *Behav. Neurosci.* *125*, 900–910.

Reviewers' Comments:

Reviewer #1:

Remarks to the Author:

The authors have incorporated all of my suggestions as well as those of the other reviewers, resulting in a significantly stronger paper which is much more accessible to the general readership.

1. The schematic explaining the modelling of the linear regression based model with a graphical guide to the predictor matrix and beta coefficients make the whole process far easier to grasp.
2. The emphasis is now on the main take home message of the paper -- highly structured social touch induced neural firing rates with a potentiation of population dynamics as nicely depicted in Fig 4.
3. The early figures do a good job of introducing the reader to the inherent diversity of social touch responses & episodes.
4. The modelling incorporating an LIF network showing inhibitory depolarization and decrease of excitatory input resistance being able to combinatorially lead to similar population dynamics is well done and the speculations regarding the role of oxytocin interesting although the authors have clearly not gone beyond speculation which is appropriate given their dataset.
5. I appreciate the use of EPSP kernels which I feel do a significantly better job with sparse neural trains than the often used Gaussian kernels.
6. I also appreciate the 'stuffed rat' data from S1, A1 and VMC which makes a strong point regarding the live conspecific dependency of social touch.

I have no further issues with the manuscript.

Best regards
Shubo Chakrabarti

Reviewer #2:

Remarks to the Author:

No further issues the authors have gone beyond what is expected, a well done study.

Reviewer #3:

Remarks to the Author:

the authors have greatly improved the manuscript. The additional methods figure and additional analyses provided have answered my concerns.

Reviewer #1 (Remarks to the Author):

The authors have incorporated all of my suggestions as well as those of the other reviewers, resulting in a significantly stronger paper which is much more accessible to the general readership.

- 1. The schematic explaining the modelling of the linear regression based model with a graphical guide to the predictor matrix and beta coefficients make the whole process far easier to grasp.*
- 2. The emphasis is now on the main take home message of the paper -- highly structured social touch induced neural firing rates with a potentiation of population dynamics as nicely depicted in Fig 4.*
- 3. The early figures do a good job of introducing the reader to the inherent diversity of social touch responses & episodes.*
- 4. The modelling incorporating an LIF network showing inhibitory depolarization and decrease of excitatory input resistance being able to combinatorially lead to similar population dynamics is well done and the speculations regarding the role of oxytocin interesting although the authors have clearly not gone beyond speculation which is appropriate given their dataset.*
- 5. I appreciate the use of EPSP kernels which I feel do a significantly better job with sparse neural trains than the often used Gaussian kernels.*
- 6. I also appreciate the 'stuffed rat' data from S1, A1 and VMC which makes a strong point regarding the live conspecific dependency of social touch.*

I have no further issues with the manuscript.

*Best regards
Shubo Chakrabarti*

Dear Shubo – Thank you for your constructive suggestions throughout the review process and your positive comments.

Reviewer #2 (Remarks to the Author):

No further issues the authors have gone beyond what is expected, a well done study.

We thank the reviewer for the positive comments and constructive suggestions throughout the review process.

Reviewer #3 (Remarks to the Author):

the authors have greatly improved the manuscript. The additional methods figure and additional analyses provided have answered my concerns.

We thank the reviewer for the positive comments and constructive suggestions throughout the review process.